# *RN7SK* small nuclear RNA controls bidirectional transcription of highly expressed gene pairs in skin

Roberto Bandiera[1], Rebecca E. Wagner[2], Thiago Britto-Borges [3], Christoph Dieterich[3], Sabine Dietmann[4], Susanne Bornelöv [5✉] & Michaela Frye [2✉]

Pausing of RNA polymerase II (Pol II) close to promoters is a common regulatory step in RNA synthesis, and is coordinated by a ribonucleoprotein complex scaffolded by the noncoding RNA *RN7SK*. The function of *RN7SK*-regulated gene transcription in adult tissue homoeostasis is currently unknown. Here, we deplete *RN7SK* during mouse and human epidermal stem cell differentiation. Unexpectedly, loss of this small nuclear RNA specifically reduces transcription of numerous cell cycle regulators leading to cell cycle exit and differentiation. Mechanistically, we show that *RN7SK* is required for efficient transcription of highly expressed gene pairs with bidirectional promoters, which in the epidermis co-regulated cell cycle and chromosome organization. The reduction in transcription involves impaired splicing and RNA decay, but occurs in the absence of chromatin remodelling at promoters and putative enhancers. Thus, *RN7SK* is directly required for efficient Pol II transcription of highly transcribed bidirectional gene pairs, and thereby exerts tissue-specific functions, such as maintaining a cycling cell population in the epidermis.

[1] Department of Genetics, University of Cambridge, Downing Street, Cambridge CB2 3EH, UK. [2] German Cancer Research Center—Deutsches Krebsforschungszentrum (DKFZ), Im Neuenheimer Feld 280, 69120 Heidelberg, Germany. [3] University Hospital Heidelberg, German Center for Cardiovascular Research (DZHK), Im Neuenheimer Feld 669, 69120 Heidelberg, Germany. [4] Washington University School of Medicine in St. Louis, 660S. Euclid Ave, St. Louis, MO 63110, USA. [5] Wellcome–MRC Cambridge Stem Cell Institute, University of Cambridge, Puddicombe Way, Cambridge CB2 0AW, UK. ✉email: smb208@cam.ac.uk; M.Frye@dkfz.de

Regulation of transcription is one of the most important steps in gene expression to ensure coordinated cellular behaviours and fate decisions. Transcription of all protein-coding genes and many noncoding RNAs is carried out by the RNA polymerase II (Pol II) complex[1]. To initiate transcription and maintain elongation, the Pol II complex interacts with a multitude of proteins and protein complexes[1]. Pol II often pauses shortly downstream of transcription initiation sites before beginning productive elongation to ensure proper 5′ capping of nascent RNAs[2,3], to prevent transcriptional re-initiation by another Pol II enzyme[4,5], and to maintain a nucleosome-free promoter[6,7]. The paused elongation complex is stabilized by the negative elongation factor (NELF) and DRB sensitivity inducing factor (DSIF)[8–10]. Release of paused Pol II into productive elongation requires the P-TEFb kinase complex and phosphorylation of serine 2 of the Pol II C-terminal domain (CTD)[11–15].

The activity of P-TEFb itself is regulated by a ribonucleoprotein complex containing RN7SK, a highly structured, abundant non-coding RNA of 331 nucleotide length[16]. RN7SK is stabilized by methyl phosphate capping enzyme (MePCE) at the 5′ end and La-related protein 7 (LARP7) at the 3′ end[17,18]. In the nucleus, RN7SK regulates transcription by sequestering P-TEFb[19,20], which is expected to prolong Pol II pausing. Pol II pausing is a common feature of gene regulation during development and embryonic stem cell differentiation[21–24], and orchestrates rapid and dynamic changes in transcription, in particular of regulators involved in signal transduction[24–27]. Despite the essential roles of Pol II pausing during development and diseases[28], the underlying molecular roles of RN7SK in regulating the transcriptional processes in adult tissue homoeostasis remains largely unexplored. Depletion experiments in mouse embryonic stem cells revealed impaired neuronal differentiation[29], and identified RN7SK as regulator of bidirectionally transcribed enhancers and transcription termination[30,31]. However, the direct transcriptional functions of RN7SK during cell differentiation have yet to be identified.

Here, we characterized the functional role of RN7SK in adult tissues using the mammalian epidermis, one of the best-characterized epithelial tissues[32]. Depletion of RN7SK triggered terminal differentiation through transcriptional repression of cell cycle regulators causing cell cycle arrest. The downregulation of genes was independent of chromatin changes in promoters and enhancers. Instead, gene repression occurred at specific highly expressed genes bidirectionally transcribed from exceedingly open, accessible promoters. Thus, our work identifies a functional role of the 7SK snRNP complex in regulating RNA synthesis during cellular differentiation in adult skin.

## Results

***Rn7sk* regulates epidermal cellularity**. To investigate the function of 7SK in gene transcription, we generated two transgenic mouse lines carrying floxed *Rn7sk* alleles (Fig. 1a). We either targeted the *Rn7sk* gene including the TATA-box (Fig. 1a, Line 1) or including the TATA-box and the proximal sequence element (PSE) in the RNA Pol III promoter (Fig. 1a, Line 2). To remove *Rn7sk* in the interfollicular epidermis (IFE) (*Rn7sk* cKO), the mice were crossed to a transgenic line carrying an inducible oestrogen receptor domain under the control of the keratin 14 promoter (*Krt14:Cre-ERT2*), targeting all undifferentiated basal cells in the IFE. To visualize recombined cells, we included a reporter transgene (*Rosa26:TdTomato*) (Fig. 1b).

Administration of 4-hydroxytamoxifen (4-OHT) for 2 weeks efficiently removed *Rn7sk* in the epidermis (Fig. 1c, d; Fig. S1a−c, upper panels). After two weeks, we measured a significant loss of cellularity in the *Rn7sk* cKO epidermis (Fig. 1d, middle and lower panels and Fig. 1e; Fig. S1d). To replace lost epidermal cells, skin often induces wound healing processes. Indeed, we measured increased proliferation and upregulation of the injury marker keratin 6 (K6) after four weeks of 4-OHT treatment (Fig. 1f; Fig. S1d−g). We observed the same phenotype when we deleted *Rn7sk* during skin morphogenesis starting from postnatal day 4 (P4) (Fig. 1g−i). Unexpectedly, one month after the last 4-OHT application, the skin returned to normal despite the complete absence of *Rn7sk* in the epidermis (Fig. S1h, i).

To test whether *Rn7sk* was essential for skin development, we generated *Rn7sk* knockout mice by crossing animals to a *Sox2:Cre* transgenic line[33], in which Cre recombinase is expressed in the inner cell mass, leading to total *Rn7sk*-deletion. Complete absence of *Rn7sk* was sublethal (Fig. S1j), but the surviving offspring was phenotypically normal, including the skin morphology (Fig. S1k).

We concluded, that acute deletion of *Rn7sk* in mice reduced epidermal cellularity in the short term, yet this effect was compensated in vivo via a mechanism that resembled a wound healing process (Fig. 1j).

**Ablation of *RN7SK* enhances epidermal cell differentiation**. To confirm that the reduction in epidermal cellularity was a direct consequence of *RN7SK*-deletion, we repressed *RN7SK* in primary human keratinocytes, a well-characterized in vitro model for IFE cells[34]. To reduce *RN7SK* expression, we used three different siRNAs targeting *RN7SK* and one against *LARP7* (Fig. S2a−c). LARP7 is essential for stabilizing *RN7SK*[17]. Of all siRNAs tested, siRNA5 reduced *RN7SK* most efficiently and was used throughout this study (Fig. S2b).

To evaluate the cellular effects of reduced *RN7SK* levels in human epidermal cells. We performed colony-forming assays and skin reconstitution tests on de-epidermized dermis[35] (Fig. 2a–c). The colony-forming efficiency of *RN7SK*-depleted cells was significantly reduced (Fig. 2a, b), and their ability to reconstitute a multi-stratified epithelium ex vivo was abolished (Fig. 2c). These results are in line with the loss of cellularity observed in our *RN7SK* cKO mouse epidermis after 2 weeks of 4-OHT treatments (Fig. 1d, e).

Next, we analyzed the consequences of *RN7SK*-depletion on keratinocyte differentiation. We induced the terminal differentiation programme by calcium induction, the best-studied pro-differentiating stimulus for keratinocytes (Fig. 2d)[36]. While expression of the undifferentiation marker *ITGA6* decreased two-fold, terminal differentiation markers (*INV*, *TGM1*) increased more than five-fold in absence of *RN7SK* (Fig. 2e, f). *RN7SK*-depletion also phenotypically enhanced terminal differentiation by inducing the formation of a stratified epithelium in culture (Fig. S2d, e). The increase in differentiation was confirmed using three different *RN7SK* siRNAs and a siRNA targeting *LARP7* (Fig. 2g). We concluded that deletion of *RN7SK* caused epidermal cell differentiation.

**RN7SK maintains robust transcription of highly expressed genes**. To understand the underlying molecular mechanisms leading to epidermal differentiation, we first tested how the ablation of *RN7SK* affected global transcription. ChIP-sequencing experiments confirmed a slight but significantly lower RNA Pol II occupancy at transcription start sites (TSS) after *RN7SK* knockdown (Fig. 3a). Calculation of the pausing index (number of reads +250 base pairs around the TSS divided by the number of reads across the rest of the gene body) indicated that ablation of *RN7SK* moved Pol II from initiation into elongation (Fig. 3b). A reduction of Pol II pausing was further confirmed by a two-fold increase in serine 2 phosphorylation levels at its C-terminal domain (CTD) in the absence of *RN7SK* (Fig. 3c, Ser2), which is

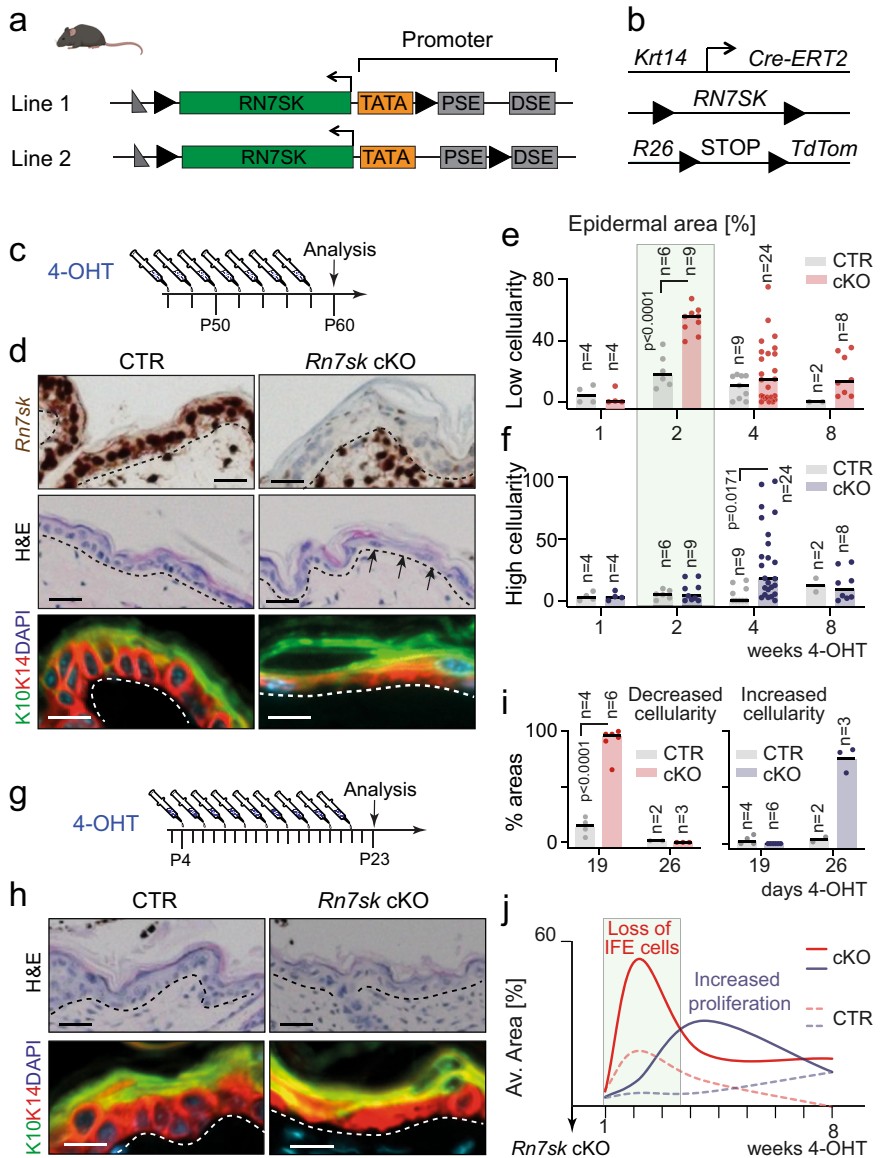

**Fig. 1 Rn7sk regulates cellularity in the mouse epidermis. a** Schematic representation of the targeted *Rn7sk* alleles in the two mouse lines generated. loxP sites = black triangles; TATA = TATA box; PSE = proximal sequence elements; DSE = distal sequence elements. **b** Schematic representation of the transgenes used to delete *Rn7sk* within the interfollicular epidermis (IFE). **c** Treatment regime of the experiment shown in (**d**). **d** *Rn7sk* RNA in situ hybridization (brown; top panel), haematoxylin and eosin staining (H&E, middle panel) and KRT10 (K10; green) and KRT14 (K14; red) immunofluorescence (bottom panel) of control (CTR) and *Rn7sk* knock-out (*Rn7sk* cKO) mice. **e**, **f** Quantification of skin area with reduced (red) (**e**) or increased (blue) (**f**) epidermal cellularity at the indicated time points (*n* = mice). **g** Treatment regime of the experiment shown in (**h**, **i**). **h** H&E staining (top panel) and KRT10 (K10; green) and KRT14 (K14; red) immunofluorescence (bottom panel) of mice treated with 4-hydroxytamoxifen (4-OHT) as shown in (**g**). DAPI (blue): nuclear counterstain (**d**, **h**). **i** Quantification of skin area with decreased (red; left panel) or increased (blue; right panel) epidermal cellularity in mice treated as shown in (**g**) (*n* = mice). **J** Graphical summary of the data shown in (**e**, **f**, **i**). Mouse line 1 was used in (**d**, **h**, **i**). Pooled data from mouse lines 1 and 2 are shown in (**e**, **f**). Scale bar: 10 μm (**d**, **h**). Shown is mean. Multiple two-tailed unpaired t-tests. Exact *p*-values are indicated. Source data are provided as a Source Data file.

required for productive elongation[37]. In contrast, transcription initiation requires phosphorylation at serine 5, and we measured no differences in *RN7SK*-depleted cells (Fig. 3c, Ser5). We concluded that depletion of *RN7SK* removed Pol II from the transcriptional start sites.

If loss of *RN7SK* decreased Pol II pausing, RNA synthesis should be enhanced. However, when we profiled newly transcribed metabolically labelled RNA (4SU-seq) (Fig. S3a)[38], we unexpectedly measured a global decrease of nascent RNAs at TSSs in the absence of *RN7SK* (Fig. 3d). This decrease of 4SU-labelled RNAs was driven by robust gene-specific repression of

highly transcribed genes (Fig. 3e, f). In contrast, the overall nascent transcript levels remained unchanged (Fig. S3b). Counterintuitively, both up- and downregulated genes exhibited a reduction in the pausing index when RN7SK was depleted (Fig. S3c, d).

To confirm that loss of *RN7SK* inhibited rather than induced transcription, we first identified all common differentially transcribed genes in two independent 4SU RNA-sequencing datasets (Fig. 3g; Fig. S3e−g). Then, we asked whether the differences in new transcription were also found at mature mRNA levels (Fig. 3h). Downregulation of nascent transcripts

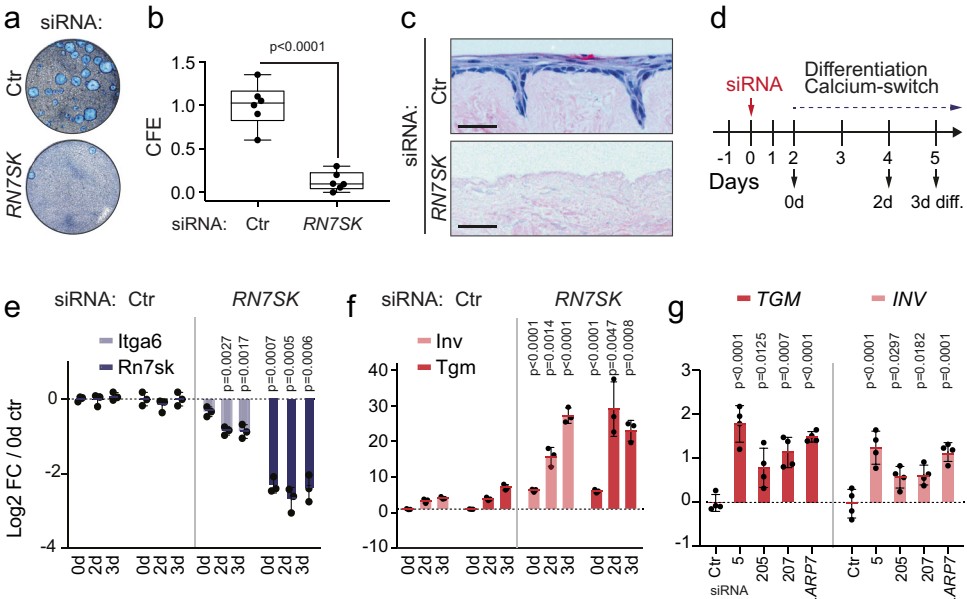

**Fig. 2 RN7SK-depletion induces differentiation of primary human keratinocytes. a**, **b** Representative image (**a**) and quantification (**b**) of the colony-forming assay (CFE) using primary human keratinocytes transfected with control (Ctr) or *RN7SK* siRNAs (*n* = 6 transfections). Box plots show all data points, median, interquartile range, and whiskers ending at minimum and maximum values. **c** H&E staining of human skin reconstituted ex vivo on de-epidermalised dermis (DED) using primary human keratinocytes transfected with Ctr or *RN7SK* siRNAs. Shown is one representative image out of two DED assays. Scale bars: 10 μm. **d**–**g** Treatment regime in days (**d**) and log$_2$ fold-change (FC) of RNA levels of Integrin α6 (*ITGA6*; light blue) and *RN7SK* (dark blue) (**e**), or Transglutaminase 1 (*TGM*; red) and Involucrin (*INV*; pink) (**f**) in transfected human keratinocytes. (*n* = 3 transfections). **g** Log2 FC of Tgm and Inv RNA levels in human keratinocytes 48 h after transfection of the indicated siRNAs (*n* = 4 transfections). 5, 205, and 207: different siRNAs targeting *RN7SK*. Data are normalized to *GAPDH* (**e**–**g**). Shown is mean ± SD (**e**–**g**). Two-tailed unpaired t-test (**b**). One-way ANOVA (**e**–**g**). Exact *p*-values are indicated. Source data are provided as a Source Data file.

correlated with reduced levels of total RNA after 24 and 48 h of *RN7SK* knockdown (Fig. 3h, lower panel). In contrast, upregulated nascent transcripts only modestly increased on total RNA level (Fig. 3h, upper panel). We concluded that *RN7SK* was required for efficient transcription of a specific set of highly expressed genes.

**7SK-sensitive genes are characterized by bidirectional transcription and open chromatin.** In search of a mechanism causing the transcriptional changes in *RN7SK*-depleted cells, we first inspected the most upregulated newly transcribed RNAs (Fig. S3h, left panel). Upregulated transcripts often contained several alternative start sites (Fig. 3i, *AKAP12*, upper panel; Fig. S3i), possibly leading to the accumulation of new transcript reads over the gene body.

A noticeable feature of the most repressed transcripts was the enrichment of sequence reads upstream of the TSS, indicating antisense transcription (Fig. 3i, lower panel, *CDT1*; Fig. S3j). To test whether the downregulated transcripts were commonly bidirectionally transcribed, we quantified all antisense sequence reads. Only downregulated nascent RNAs contained a higher number of antisense transcripts when compared to all genes (Fig. 3j). Using CAGE sequencing data from the FANTOM project, we confirmed that the downregulated genes were twice as likely to have a bidirectional promoter when compared to all genes (*p* = 3e−5, Fisher's exact test) in human epidermal cells (Fig. 3k). Moreover, ATAC sequencing data from human keratinocytes[39] revealed that the corresponding promoters were highly accessible, even more so than promoters of upregulated genes (Fig. 3l, m; Fig. S3k). Thus, deletion of *RN7SK* specifically reduced transcription from bidirectional promoters of highly expressed genes marked by open and accessible chromatin.

**7SK regulates bidirectional transcription of highly expressed gene pairs.** At bidirectional promoters, Pol II initiates transcription divergently from a central promoter and undergoes promoter-proximal pausing into both directions[40–42]. We hypothesized that *RN7SK* may be required to organize the symmetry of Pol II bidirectional transcriptional initiation. To test this hypothesis, we first determined all protein-coding genes having another protein-coding gene upstream in sense (ss) or anti-sense (as) direction less than 1 kb away (Fig. 4a). As an additional control, we performed the same analysis searching for downstream genes (Fig. 4a). We discovered that 7SK-sensitive genes were about three times more likely to have an upstream anti-sense gene (3.1-fold in 4SU and 2.4-fold in total RNA sequencing datasets) (Fig. 4b). We also observed a significant enrichment of downstream genes in sense direction, albeit to a lesser extent (Fig. 4c).

Bidirectional gene pairs are often co-expressed and transcription is initiated in both directions through shared regulatory elements[43]. When we asked whether the anti-sense orientated gene pairs were co-regulated, we found a correlation for some but not all gene pairs (Fig. 4d, e). The co-regulation was more pronounced when the upstream anti-sense gene was also highly expressed (Fig. 4f). RNA levels of lower expressed upstream anti-sense genes remained largely unchanged (Fig. 4g). Bidirectional promoters often co-regulate expression of genes that function in the same biological pathway[44–46], or coordinate expression through different timepoints, such as genes involved in DNA repair and the cell cycle[47,48]. Indeed, protein-coding genes containing an upstream anti-sense gene in close proximity (<1 kb) were enriched in regulating DNA repair, cell cycle, and RNA metabolism (Fig. 4h). Notably, significantly downregulated transcripts in the absence of *RN7SK* were similarly enriched for genes involved in DNA repair and cell division (Fig. 4i). One

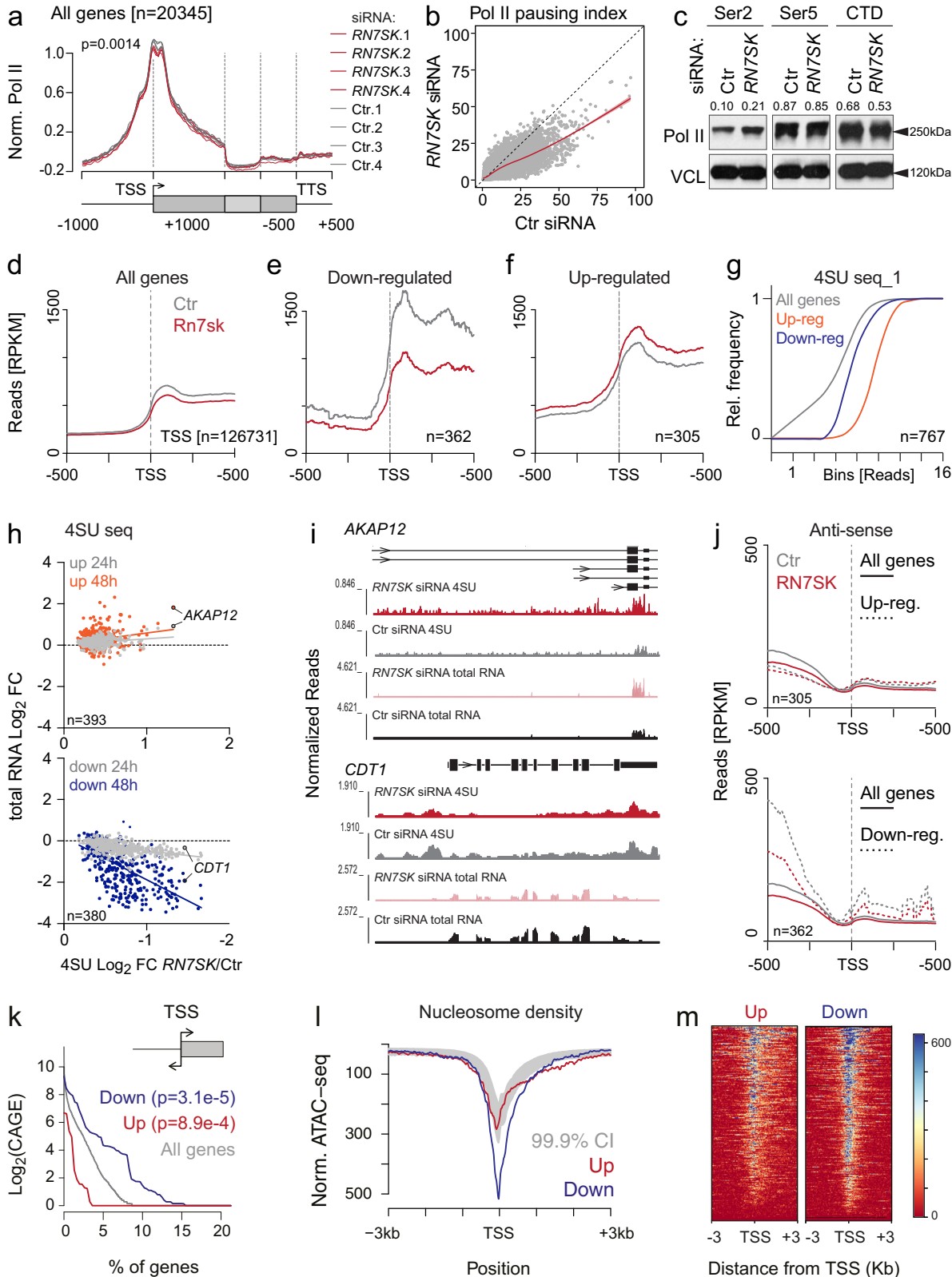

well-known gene cluster regulated by bidirectional transcription contains histone genes, where bidirectional promoters are used to maintain stoichiometry[49–51]. Accordingly, a quarter of all repressed nascent transcripts in the absence of *RN7SK* were histone genes (Fig. S4a, b). We concluded that *RN7SK* was required for efficient bidirectional co-expression of highly expressed gene pairs.

**RN7SK-mediated gene repression is not caused by changes in chromatin.** Since *RN7SK* regulated epidermal cellularity in mouse, we more closely investigated how transcription of cell cycle genes was affected. We selected consistently downregulated genes (*CDK1*, *CDC25c*, *CDC45*, and *MCM10*) and confirmed their significant reduction as early as 12 h after *RN7SK*-depletion (Fig. S4c). Upregulated genes remained largely unaffected within

**Fig. 3 RN7SK knockdown induces robust repression of highly transcribed genes. a** Metagene plot of RNA Pol II occupancy across protein-coding genes in control (Ctr) siRNA (grey) and *RN7SK* siRNA (red) transfected cells. Normalization was done using all unchanged genes in Ctr and *RN7SK* siRNA transfected cells. The first 1000 and last 500 bases of the transcript are shown unscaled, the region between has been scaled to the same length. Two-tailed unpaired t-test between Ctr and *RN7SK* signal at the TSS. **b** Plot of RNA Pol II pausing index in primary human keratinocytes transduced with Ctr or *RN7SK* siRNAs. Each dot represents one gene. **c** Western blot for RNA Pol II Phospho-Serine 2 (Ser2), Phospho-Serine 5 (Ser5), and the C-terminal domain (CTD). VCL: vinculin. Representative blot of two independent transfections. **d–f** Metagene plots of 4SU sequencing read across all genes (**d**) and significantly (padj < 0.05; Wald test with FDR correction) downregulated or upregulated new transcripts (**e, f**) around the transcription start site (TSS). **g** Cumulative relative (Rel.) frequency of sequence read counts of all genes (grey) or up- (orange) or down- (blue) regulated new transcripts. **h** Correlation of significantly (padj < 0.05; Wald test with FDR correction) up- (upper panel) or downregulated (lower panel) new transcripts with total RNA levels after 24 (grey) and 48 (up: red; down: blue) hours (h) after *RN7SK* knockdown. Highlighted examples are shown in (**i**). **i** UCSC genome browser shots of 4SU and total RNA sequencing reads of *AKAP12* (upper panel) and *CDT1* (lower panel). **j** Metagene plots of antisense reads of upregulated (upper panel) and downregulated (lower panel) 4SU sequencing reads around the transcriptional start site (TSS) compared to all genes (dotted lines) in control (Ctr) and *RN7SK*-depleted cells. n = genes in (**d–h, j**). **k** Percent of genes with antisense transcription in the first 100 nucleotides opposite to the TSS of up- (red), downregulated (blue) genes in 4SU RNA seq dataset, or all genes (grey). **l** Density plot showing ATAC seq normalized reads at transcription TSSs of up- (red) or downregulated (blue) genes in 4SU RNA seq dataset at 24 h. Grey band: 99.9% confidence interval (CI). **m** Heat maps showing ATAC seq signal at TSS of up- (top) or downregulated (bottom) genes in 4SU RNA seq (padj < 0.05; Wald test with FDR correction, Log₂Fc > 0.3). Source data are provided as a Source Data file.

24 h of *RN7SK* knock-down (Fig. S4d). The repression of cell cycle regulators in the absence of *RN7SK* was confirmed in three other independent human keratinocyte lines (Fig. S4e), and we obtained similar results when using two different *RN7SK* siRNAs (205 and 207) or a siRNA targeting *LARP7* (Fig. S4f). Because we found a slight reduction of global Pol II occupancy at transcriptional start sites, we confirmed that *GAPDH*, used for normalization, was not significantly repressed in any condition (Fig. S4g). Furthermore, co-expression of a wild-type or mutated version of human *RN7SK* (not targeted by siRNA 5) in *RN7SK* knock-down cells, prohibited efficient downregulation of cell cycle regulators (Fig. S4h, i). Since not all genes rely on P-TEFb activity for transcription[52], we also confirmed that inhibition of P-TEFb with 5,6-dichloro-1-β-D-ribofuranosylbenzimidazole (DRB) caused downregulation of the four cell cycle regulators (Fig. S4j). Thus, our data revealed that removal of *RN7SK* in primary human epidermal cells directly repressed cell cycle genes.

We next asked whether loss of *RN7SK* correlated with the formation of repressive chromatin at transcriptional start sites of these cell cycle regulator genes[6,53]. However, when we depleted *RN7SK*, nucleosome positioning around the TSS remained unaltered (Fig. 5a–d). Our results excluded chromatin remodelling at the TSS, yet high levels of *RN7SK* occupancy have also been reported at active enhancers, where it limits enhancer-RNA transcription[31]. However, ChIP-sequencing experiments for two histone modifications commonly found at putative enhancers and promoters (H3K4Me1, H3K27Ac)[54], revealed no differences in occupancies (Fig. 5e–j; Fig. S5a, b). Thus, 7SK-driven gene expression appeared to be independent of chromatin remodelling at promoters and putative enhancers.

***RN7SK* orchestrates mRNA synthesis and splicing.** Next, we investigated whether mis-regulation of bidirectional transcription was sufficient to explain the epidermal phenotype caused by *RN7SK*-depletion. We asked how the loss of *RN7SK* affected mRNA processing and degradation for two reasons. First, downregulated genes were more likely to have another downstream sense-strand gene in close proximity (<1 kb) (Fig. 4c). Second, 7SK has been described to prevent transcription downstream of polyadenylation sites[30]. To determine whether RNA synthesis, processing, or degradation was the most prevalent RNA metabolic pathways affected by depletion of *RN7SK*, we used INSPEcT, a tool that integrates intronic and exonic signals from nascent and total RNA-seq data to derive the rates of pre-mRNA synthesis, processing, and mature mRNA degradation[55,56]. In

both independent RNA sequencing datasets, RNA synthesis was most affected by depletion of *RN7SK* (Fig. 6a−d; Fig. S6a−d). Since mRNA syntheses take place in the nucleus, we confirmed the downregulation of genes in the nucleus as early as 18 h after *RN7SK* depletion (Fig. 6e).

Since some gene expression changes were predicted to be caused by RNA processing and degradation, we tested how *RN7SK*-depletion affected splicing. Indeed, we found increased intron retention levels in downregulated genes in both RNA and 4SU sequencing datasets (Fig. 6f, g). Using rMATS to identify alternative splicing events genome-wide[57], we revealed exon skipping as the most prevalent splicing differences (Fig. 6h; Fig. S6e). The splicing differences were consistent in both independent RNA sequencing datasets, but the overlapping genes were not enriched in cell cycle regulation, DNA repair, or chromosome organization (Fig. 6i−k; Fig. S6f−h). Thus, RNA synthesis and splicing were likely to be regulated by independent mechanisms involving 7SK.

**Terminal differentiation is the consequence of cell cycle arrest in the absence of *RN7SK*.** To understand the molecular mechanisms causing terminal differentiation in the absence of *RN7SK*, we transcriptionally profiled undifferentiated and calcium-differentiated epidermal cells (Fig. S7a). Several lines of evidence indicated that loss of *RN7SK*- and calcium-induced differentiation were two distinct regulatory processes. First, the number of differentially expressed genes in response to deletion of *RN7SK* was three-fold larger than transcriptional changes induced by the calcium-switch alone (Fig. S7b). Second, the differential gene expression profile of *RN7SK* knock-down cells in calcium-low and -high conditions overlapped by more than 70% (Fig. S7b). Third, while both calcium-induced differentiation and *RN7SK*-depletion increased expression of genes involved in epidermis differentiation (Fig. 7a; Fig. S7c), only *RN7SK* depletion repressed cell cycle genes (Fig. 7b; Fig. S7d).

Since cell-cycle withdrawal is an early hallmark of skin differentiation that occurs already in the undifferentiated basal epidermal compartment[58], we asked whether cell cycle arrest explained the induction of terminal differentiation in the absence of *RN7SK*. Indeed, *RN7SK*-depleted cells accumulated in G2/M phase of the cell cycle 48 h after siRNA transfection (Fig. 7c, d; Fig. S7e). Cell cycle arrest was confirmed in four independent epidermal lines, yet one line arrested in G1 (Fig. 7d; Fig. S7f). As expected, the cell cycle was also affected by calcium-induced differentiation (Fig. S7g). We further confirmed a reduction of

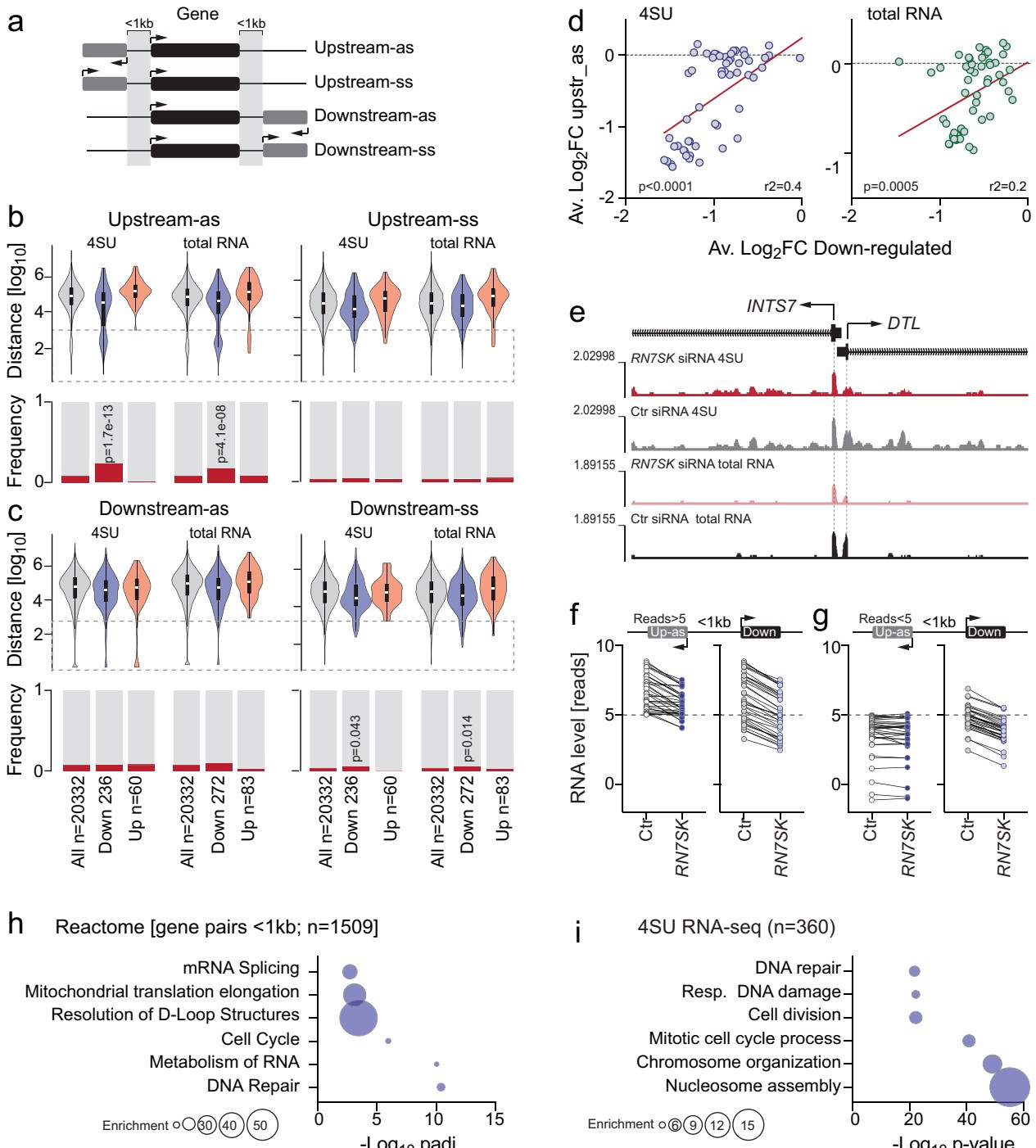

**Fig. 4 7SK orchestrates bi-directional transcription of highly expressed gene pairs. a** Illustration of the analyses shown in (**b**, **c**). **b**, **c** Distance (upper panels) and frequency (lower panels) of genes with upstream (**b**) or downstream (**c**) antisense (as) (left panels) or sense (ss) (right panels) genes within 1000 bases (1 kb) distance (grey dotted box). Up- (orange) and down- (blue) regulated genes were defined based on two independent 4SU and total RNA-seq datasets. All (grey) = all protein-coding genes; Up = genes with log2FC > 0.3 and padj < 0.05; Down = genes with log2FC < −0.3 and padj < 0.05 (Wald test with FDR correction). The Up and Down groups were compared to all protein-coding genes. Violin plots show the median and interquartile range and width indicating frequency (**b**, **c** upper panels). *P*-value calculation (see "Methods"). **d** Correlation of RNA levels of downregulated genes with their upstream as genes shown as average log2 fold-changes (FC) in two independent 4SU (left panel) and total RNA (right panel) sequencing datasets. *P*-value tests slope deviation from 0. **e**–**g** UCSC genome browser shots of bi-directional repressed *DTL* and its upstream anti-sense gene partner *INTS7* (**e**). RNA levels (reads) of new transcripts of downregulated genes (padj < 0.05; FC < −0.5; *n* = 485) with bi-directional as gene less than 1 kb away and more than an average of 5 reads (*n* = 36 genes) (**f**) or less than 5 reads (*n* = 33 genes) (**g**) in upstream antisense (dark blue) or downstream sense (light blue) directions in control (ctr) siRNA transduced cells. **h** Reactome of all bi-directionally orientated gene pairs less than 1 kb apart. **i** Gene ontology analyses using all common significantly (padj < 0.5; Wald test with FDR correction) downregulated genes in the 4SU RNA seq datasets. Background: all expressed genes. Source data are provided as a Source Data file.

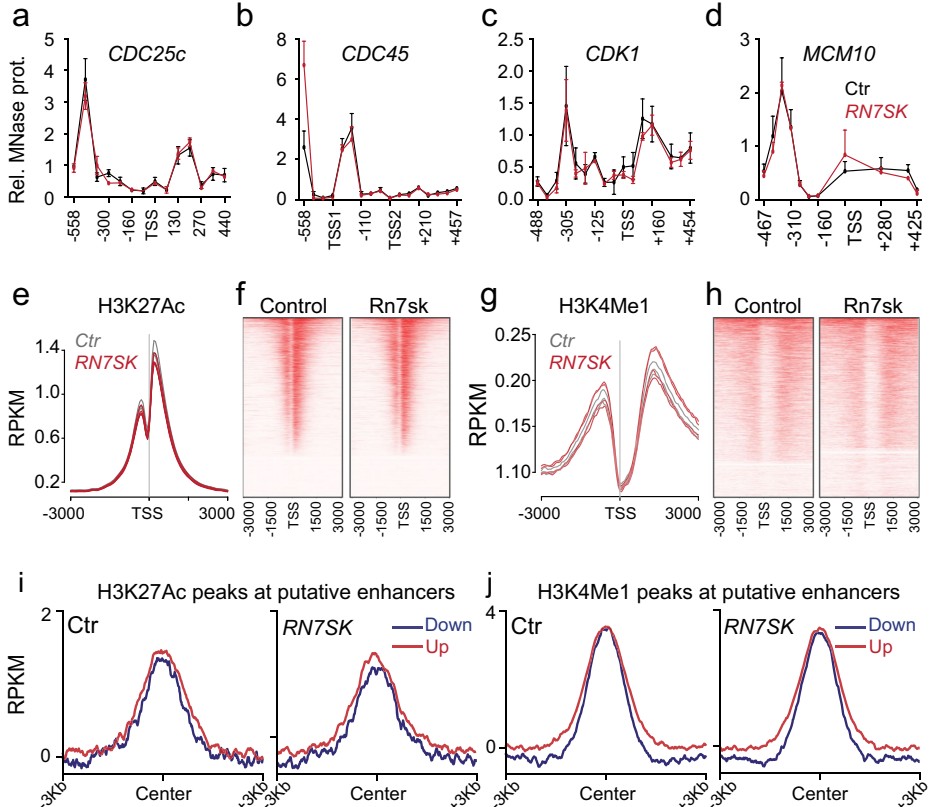

**Fig. 5 Chromatin at enhancers and promoters is unaltered in RN7SK knock-down cells. a–d** Micrococcal Nuclease (MNase) protection assay around the transcriptional start sites (TSS) of four downregulated candidate genes showing nucleosome distribution in control (Ctr) (black lines) and *RN7SK* knock-down (red lines) cells 24 h after siRNA transfection (n = 4 transfections). Data are presented as mean values ± SD. **e–h** Occurrence of H3K27Ac (**e**) and H3K4Me1 (**g**) ±3000 bases from the TSS in Ctr (grey) or *RN7SK* knock-down (red) cells (n = 4 ChIP-seq experiments). Representative heatmaps (**f, g**) showing sequencing reads from data in (**e, g**). **i, j** Distribution of H3K27Ac (**i**) or H3K4Me1 (**j**) ChIP seq signal ±3000 bases from the enhancer centre in up- (red) or downregulated (blue) genes in 4SU RNAseq dataset (padj < 0.05, Log₂Fc > 0.3 or < −0.3) in Ctr (left hand panels) or *RN7SK* knock-down (right hand panels) cells. Shown is one representative ChIP-seq experiment out of four replicates. Source data are provided as a Source Data file.

cell cycle genes in *RN7SK* cKO mouse epidermis (Fig. 7e). We concluded that downregulation of cell cycle genes leading to cell cycle arrest induced terminal differentiation of *RN7SK*-lacking epidermal cells.

To prove that the cell cycle changes were directly caused by loss of *RN7SK*, we re-expressed a wild-type or mutated *RN7SK* construct in a *RN7SK*-depleted human epithelial cancer line (FaDu) (Fig. 7f). As expected, cell cycle regulators were downregulated in the absence of *RN7SK* (Fig. S7h). Yet, re-expression of wild-type and mutated *RN7SK* in these depleted cells prohibited the accumulation of cells in the G2/M phase of the cell cycle (Fig. 7g). We concluded that epidermal cells required *RN7SK* to coordinate the expression of cell cycle genes, and disruption of this co-regulation stimulated terminal differentiation processes through cell cycle exit.

In summary, our data demonstrated that the 7SK snRNP complex orchestrates efficient transcription of highly expressed bidirectionally transcribed gene pairs potentially by tethering Pol II to the transcriptional start sites. Loss of *RN7SK* in the epidermis specifically represses cell cycle genes causing cell cycle arrest and thereby stimulates differentiation, a process that was reversible in vivo.

## Discussion

Here, we investigated the transcriptional roles of *Rn7sk* in adult tissues, using the epidermis as a model system. We show that human *RN7SK* sustained Pol II activity at highly expressed

bidirectionally transcribed gene pairs. Although the 7SK ribonucleoprotein complex regulates P-TEFb activity, we find no evidence that *RN7SK*-depletion enhances RNA synthesis due to an increased release of Pol II. However, previous studies identified an inhibitory effect of 7SK on P-TEFb in response to stress such as ultraviolet radiation[19,20], while uninduced cells showed little changes in global transcription upon *RN7SK*-depletion[59]. Thus, the release of P-TEFb from 7SK upon stress might rather reflect the transcriptional reprogramming upon the stress signal than a general inhibitory role of the complex on transcriptional elongation.

Our finding that *RN7SK* was required to maintain robust bidirectional transcription of highly expressed gene pairs implies a structural role for 7SK at promoters with complex and high turnover of Pol II. An unexpectedly high RNA Pol II turnover has also been reported at paused promoters[60]. We propose that the 7SK ribonuclear complex tethers P-TEFb and other transcriptional regulators to highly transcribed bidirectional promoters to regulate Pol II activity. Our data confirm that bidirectionally transcribed genes often regulate DNA repair, cell cycle, and RNA metabolism, a highly efficient way of coordinating the expression of genes acting in the same cellular response pathway[44–48]. For instance, out of 120 examined human DNA repair genes, 42% are arranged in a bidirectionally divergent configuration with transcription start sites less than 1 kb apart[48].

In epidermal cells, cell cycle regulators and histones genes were amongst the most sensitive genes to *RN7SK*-depletion. Histone genes

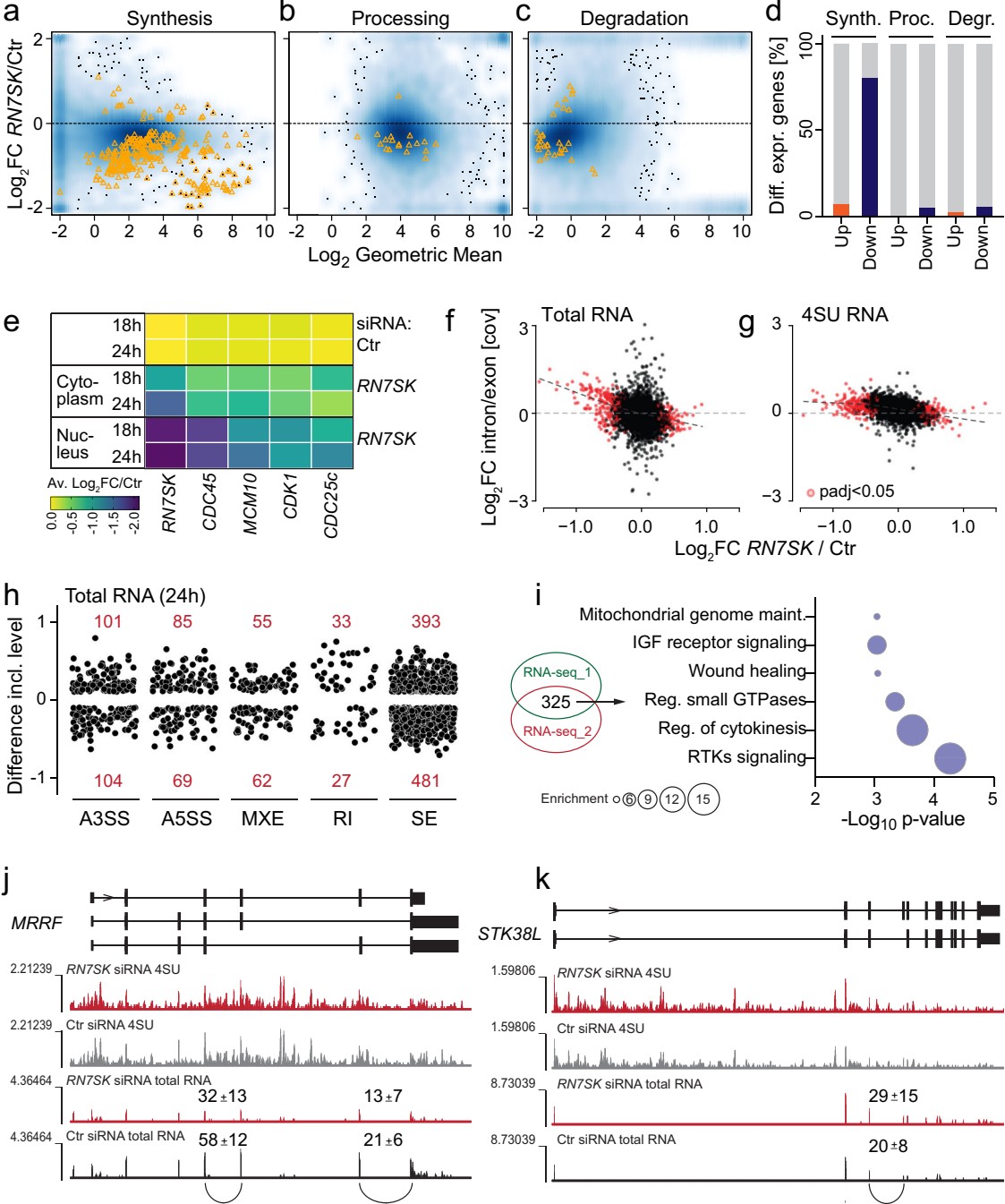

**Fig. 6 Loss of RN7SK primarily affects RNA synthesis. a–c** Density scatter plots (darker colours for higher density) of changes in the absence of *RN7SK* versus time points (0 and 24 h and 10 min RNA labelling time). Shown are changes in RNA levels caused by differences in RNA synthesis (**a**), processing (**b**), and mature mRNA degradation (**c**). Each yellow triangle is one gene. **d** Percent of differentially expressed genes shown in (**a–c**). **e** Average (Av.) log₂ fold-change (FC) of RNA levels of the indicated genes in the cytoplasm or nucleus after 18 and 24 hours (h) knock-down of *RN7SK* measured by RT-qPCR. (*n* = 4 transfections). **f, g** Correlation of log₂ (Fc) expression and log₂ (intron/exon coverage) in total (**f**) or 4SU (**g**) RNA seq datasets 24 h after *RN7SK* siRNA transfection. **h** Splicing differences in total RNA 24 h after *RN7SK*-depletion (splicing FDR > 0.05; inclusion difference >0.1 or < −0.1). A3SS: alternative 3′ splice site; A5SS: alternative 5′ splice site; MXE: mutual exclusive exon; RI: intron retention; SE: exon skipping. **i** Gene Ontology analysis using differentially spliced transcripts overlapping in two independent total RNA-seq datasets. **j, k** Examples of alternatively spliced transcripts resulting in down- (**j**) or up- (**k**) regulation of the RNA shown as UCSC genome. Source data are provided as a Source Data file.

are bidirectionally transcribed to maintain stoichiometry[49–51]. As a consequence, depletion of *RN7SK* induced epidermal cells to exit the cell cycle and undergo differentiation. However, in vivo, this loss of cellularity was later compensated by a wound-like response. The different phenotype in response to *RN7SK*-depletion in vitro versus in vivo can be explained by our finding that *RN7SK* affects transcription gene-specifically, thereby regulating cell context-specific

functions. For instance, in response to acute deletion of *Rn7sk* in mouse skin, cycling epidermal populations will trigger the terminal differentiation programme due to a synchronized cell cycle exit. However, the cell cycle regulators are still expressed, albeit with lower levels. The overall reduction of cycling might lead to a slightly lower, yet sustainable, epidermal turn-over. In contrast, deletion of *Rn7sk* in mouse embryonic stem cells specifically repressed a different

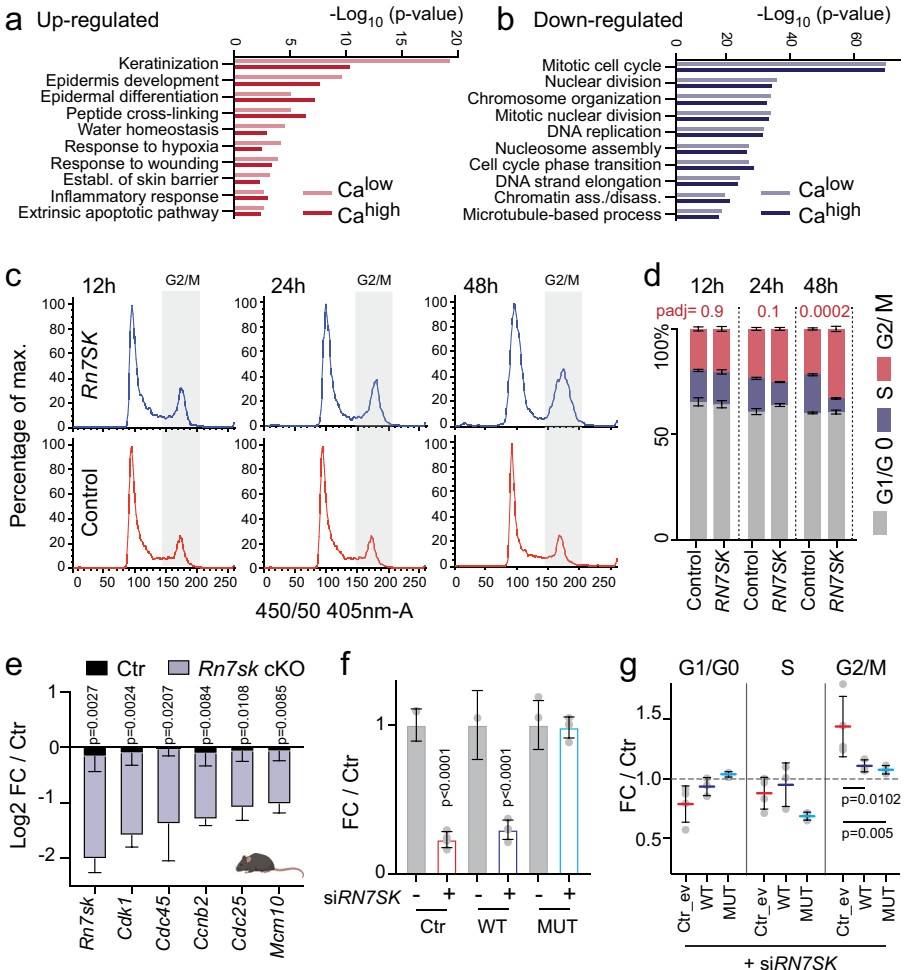

**Fig. 7 Cell cycle exit causes differentiation in RN7SK-depleted epidermal cells. a**, **b** Gene ontology analysis of up- (red; **a**) or downregulated (blue; **b**) genes (padj < 0.001; log$_2$FC > or < 0.75) in response to *RN7SK* knock-down in human keratinocytes grown in calcium low (Ca$^{low}$; light bars) or high (Ca$^{high}$; dark bars) conditions. **c**, **d** Cell cycle profiles of primary human keratinocytes (**c**) and quantification of percent of cells in the different phases of cell cycle (**d**) after 12, 24, or 48 h in control or *RN7SK* siRNA treated cells (*n* = 5 transfections). Data are presented as mean values ±SD. **e** RT-qPCR for five cell cycle regulators in mouse epidermis of control (black) or *RN7SK* cKO (blue; mouse line 1) animals treated for 2 weeks with 4-OHT. **f** *RN7SK* levels after repression (si*RN7SK*) in control empty vector (Ctr; red border) transfected cells or cells over-expressing wild-type (WT; dark blue border) or mutated (MUT; light blue border) *RN7SK* constructs in an epidermal cancer line (FaDu) (*n* = 4 transfections). **g** Quantification of *RN7SK*-depleted cells in G1/G0-, G2/M- and S-phase of the cell cycle in the presence of an empty vector as control (Ctr_ev; red) or the wild-type (WT; dark blue) or mutated (MUT; light blue) *RN7SK* construct. (*n* = 4 transfections). Shown is mean ±SD (**d**–**g**). Two-way ANOVA (multiple comparisons) (**d**, **f**, **g**). Multiple two-tailed unpaired t-tests (**e**). Exact *p*-values are indicated. Source data are provided as a Source Data file.

cohort of transcriptionally poised genes with bivalent or activating chromatin marks[30]. Embryonic stem cells display unique cell-cycle features with a prolonged S-phase but truncated G1 and G2 phases[61].

Although chromatin changes are known to influence Pol II pausing[62,63], we detected no epigenetic changes at promoters or enhancers in the absence of *RN7SK*. However, we cannot exclude a faster transcription elongation rate due to the absence of tightly controlled Pol II activity, which might cause impaired splicing leading to RNA decay. Increased RNA Pol II elongation rates can affect co-transcriptional splicing and splicing efficiency, which then compromises splicing fidelity[64].

In summary, our work demonstrates that the precise co-ordination of highly expressed bidirectional gene pairs required the 7SK ribonuclear complex for epidermal homoeostasis.

## Methods
**Mice**. All mice were housed in the Wellcome Trust-Medical Research Council Cambridge Stem Cell Institute Animal Unit. All mouse husbandry and experiments were carried out in compliance with the Animals (Scientific Procedures) Act 1986

following ethical review and approval by the University of Cambridge Animal Welfare and Ethical Review Body (AWERB) under the terms and conditions of the UK Home Office licences PPL80/2619 and PPLP36B3A804.

**Generation of the *Rn7sk* cKO transgenic lines**. All transgenic lines were bred on a mixed background (F1 of B6SJL x CBA). To generate *Rn7sk* conditional knockout mice (cKO), we produced targeting vectors by BAC recombineering[65]. To generate the homology arms BACS bMQ337m02 and bMQ215m24 (Source Bioscience) were used. We generated two targeting vectors, which differ for the position of the *loxP* site located at the 5′ end of the *Rn7sk* gene locus, whereas the 3′ end *loxP* site was in the same position in both constructs. In line 1, the 5′ *loxP* site was located between the TATA Box and the proximal sequence element (TATA and PSE, respectively) of the *Rn7sk* promoter (RNA Pol II promoter). In line 2, the 5′ *loxP* site was instead positioned between the PSE and the distal sequence element (DSE). Both lines showed the same phenotype. If not otherwise stated, all displayed data were obtained from line 1.

E14 mouse embryonic stem cells were targeted by homologous recombination with the *Rn7sk* cKO targeting vector. After 10 days in the selection medium containing G418, around 100 clones per construct were picked and screened by PCR for integration of both 3′ and 5′ arms. Three double-positive clones per construct were injected into blastocysts derived from C57BL6 females. Chimeric offspring was mated with C57BL6 mice and agouti F1s were genotyped for the presence of the targeted *Rn7sk* allele. One line per construct was further mated with

FLP mice to induce excision of the selection cassette to generate the *Rn7sk* $^{flox/flox}$ mice. Mouse genotyping was performed by standard PCR (Supplementary Table 1).

Generation of *Rn7sk*-knockout mice was achieved by mating *Rn7sk*$^{/flox}$ with *Sox2:Cre* mice[33] To achieve interfollicular epidermis (IFE) specific deletion of *Rn7sk*, *Rn7sk*$^{/flox}$ mice were further crossed to *Tg(KRT1-cre/ERT)20Efu/J* mice (*Krt14:Cre-ERT* in the text)[66] and *Gt(ROSA)26Sor*$^{tm9(CAG-tdTomato)Hze}$ (*Rosa26:TdTomato*)[67]. To conditionally delete the *Rn7sk* gene in the IFE, the back skin of mice was topically treated every second day for the indicated time with 200 µl of a 4-hydroxy-tamoxifen (Sigma Aldrich) solution of 14,28 mg/ml in acetone.

**Tissue processing and staining.** The mouse epidermis was fixed overnight in 4% paraformaldehyde, dehydrated in ethanol gradient followed by xylene, paraffin-embedded, and then cut using a microtome into 5 µm thick sections. Before staining, sections were deparaffinised in xylene and rehydrated in an ethanol gradient. For immunostaining, antigen retrieval was performed by boiling the sections for 20 min in sodium citrate solution, followed by blocking in 10% FBS in PBS tween. Primary antibodies (Supplementary Table 2) were diluted in blocking solution and incubated overnight at 4 °C. After washing three times in PBS, sections were incubated with the appropriate secondary antibody (Alexa fluor-conjugated; Thermo Fisher Scientific) diluted 1:500 in PBS tween and DAPI when indicated. After washing slides were then mounted in a 1:1 solution of PBS and Glycerol. For RNA in situ hybridization of *Rn7sk*, an RNAscope probe (ACD bio) was used following manufacturer instructions. Images were acquired with an Axioplan2 microscope (Zeiss).

**Quantification of epidermal cellularity.** Epidermal cellularity was quantified on images of large areas of HE stained sections using ImageJ (FIJI). Epidermal cellularity was considered normal when a continuous single layer of nuclei was following each other; reduced if gaps were present between nuclei, increased if more than one layer of nuclei was present. Data are represented as the percentage of the skin surface that fell into each category.

**Quantification of epidermal proliferation.** To quantify IFE proliferation, sections of mouse skin were stained with Ki67 antibody as described above. Images were processed with Volocity (Perkin Elmer). The entire IFE or only the basal layer was manually selected for analysis. The software was asked to identify nuclei on the DAPI channel and then to identify how many of those had also a positive signal in the Ki67 channel to identify the percentage of Ki67+ cells.

**Cell culture, transfection, and infection.** Unless otherwise stated, neonatal primary human keratinocytes (ScienCell; #2100) were cultured in KGM gold (Lonza) or EpiLife added with human keratinocytes growth supplements (Thermo Fisher) in presence of 0.06 mM CaCl$_2$ and without the addition of antibiotics, in a humidified incubator at 37 °C with 7% CO$_2$. To induce differentiation, the calcium concentration was raised to 1.2 mM. siRNAs were transfected using Lipofectamine RNAi Max (Invitrogen), following manufacturer instruction using 1 µl of Lipofectamine for each 0.6 µl of 10 µM siRNA. siRNAs were added to cells at a final concentration of 10 nM (Supplementary Table 3). For P-TEFb inhibition, cells were transfected with either control or *RN7SK* siRNAs and treated with 50 µM 5,6-Dichlorobenzimidazole 1-β-D-ribofuranoside (DRB) (Sigma Aldrich) for the last 6 h.

FaDu (HTB-43; ATCC) cells were grown in Eagles minimum essential medium (ATCC) in the presence of Penicillin Streptomycin and 10% FBS, in a humidified incubator at 37 °C with 5% CO$_2$. siRNAs were transfected using Lipofectamine RNAi Max (Invitrogen). The recommended concentration of lipofectamine and siRNA were used, 2 µl of Lipofectamine for each 1.2 µl of 10 µM siRNA. siRNAs were added to cells at a final concentration of 20 nM.

To over-express the wild-type or siRNA 5 resistant mutant *RN7SK* construct, the full-length RNA was cloned into pSUPER (Addgene). The mutations were generated by swapping three bases at position 226–228 from AAA to TTT and on the opposite strand at position 246–248 from TTT to AAA (Fig. S2a).

To infect the wildtype and mutated *RN7SK* constructs into FaDu cells, Phoenix A cells were cultured with DMEM (Thermo Fisher) supplemented with 10% FBS (Lif Technologies), 1% non-essential amino acids (Life Technologies), 1% glutamate (Fischer Scientific) and penicillin-streptomycin (Life Technologies). These cells were transfected with 20 µg of plasmid (either an empty plasmid, or containing wildtype *RN7SK* or *RN7SK* with a mutation preventing siRNA targeting) using CaCl$_2$ and HEPES Buffered Saline (HBS 2×) (Fischer Scientific). FaDu cells were then transduced twice and polybrene (Sigma Aldrich) was added to aid transduction. Twenty-four hours post transduction selection was started using puromycin at a final concentration of 1 µg/ml. Cells were then cultured in the presence of puromycin.

To infect human keratinocytes, the constructs were first transfected into Phoenix E and the retroviral supernatants were used to stably infect AM12 cells. Keratinocytes were infected by coculture with AM12 cells. AM12 cells were removed and keratinocytes cultured in a defined medium were transfected with siRNAs as described earlier.

**Western blot.** Proteins were extracted from a pellet of human primary keratinocytes using RIPA buffer (50 mM Tris HCl pH 8.0, 150 mM NaCl, 1% Nonidet P40, 0.5% sodium deoxycholate, 0.1% SDS) supplemented with complete protease inhibitor and PhoSTOP (Roche). 50 µg of extracts were separated on 8% polyacrylamide gels and blotted onto nitrocellulose membranes. The membranes were blocked 1 h in 5% milk in TBST buffer and incubated overnight at 4 °C with the indicated antibodies. Next day they were washed in TBST and incubated 1 h at room temperature with the appropriate HRP-conjugated secondary antibodies (1:10,000 in PBS) washed three times in TBST. The bands were visualized after incubation with ECL prime (Amersham).

**Colony-forming assay.** Keratinocytes cells were cultured in KGM gold with 0.06 mM CaCl for up to 24 h after transfection at which point they were seeded onto mitomycin C-treated (Sima-Aldrich) (4 µg/ml for 2 h) 3T3-J2 (Kerafast) mouse embryonic fibroblasts (500 cells in each 10 cm dish) and cultured in complete low-calcium FAD medium (one part of Ham's F12, three parts of DMEM, 10% FBS, 18 mM adenine, 0.05 mM calcium, 0.5 mg/ml hydrocortisone, 5 mg/ml insulin, 0.1 nM cholera enterotoxin, and 10 ng/ml epidermal growth factor) until colonies were clearly visible (about 3 weeks). The plates were fixed in 4% formaldehyde for 10 min, washed in PBS, and stained with 1% rhodamine B and 1% nile blue in PBS for 20 min. After staining, the cells were washed in water and imaged. Images were acquired with an Olympus ix 51 microscope with prior XY stage, Z drive, DP72 camera, and Cell Sens software. The colony number was manually quantified. Colony-forming assays were replicated twice.

**Skin reconstitution on de-epidermized dermis (DED).** Keratinocytes cells were cultured in KGM gold with 0.06 mM CaCl for up to 24 h after transfection at which point $4 \times 10^5$ cells were seeded onto de-epidermized human dermis (Addenbrookes Tissue Bank) and cultured in a transwell at the liquid—air interface with complete low-calcium FAD medium. About 3 weeks after seeding, the piece or reconstituted skin was fixed overnight in 4% paraformaldehyde and processed as described in the histology section. Skin reconstitution assays were replicated twice.

**RNA extraction and RT-qPCR.** RNA was extracted using TRIZOL reagent (Thermo Fisher Scientific) following manufacturer instructions and quantified using a Nanodrop. cDNA was synthesized from 1 µg of RNA using Superscript III (Thermo Fisher Scientific) and following manufacturer instructions. RT-qPCRs were performed using either Fast SYBR green master mix or Taqman fast universal master mix (Thermo Fisher Scientific). Taqman probes were purchased from Thermo Fischer Scientific (Supplementary Table 4). RT-qPCR results of target genes were normalized to *GAPDH* or 18S rRNA (Mm03928990_g1).

For nuclear and cytoplasmic RNA extraction, human primary keratinocytes were collected with trypsin, washed in PBS, centrifuged and the pellet was processed with the NE-PER kit (Thermo Fischer Scientific) following manufacturer instructions. Four replicates of each sample were processed, and the experiment was performed twice.

For quantification of *Rn7sk* levels in mouse IFE, back skin was isolated from *Rn7sk cKO* mice and was quickly snap frozen after fat was removed. Frozen skin was then homogenized in TRIZOL reagent (Thermo Fisher Scientific) and RNA was then extracted following manufacturer instruction.

**Chromatin immmunoprecipitation.** Epidermal cells were grown to 60–70% confluency and transfected with control or *RN7SK* siRNA 5 (8 × 15 cm dishes per siRNA). 18 h after transfection cells were fixed with 1% formaldehyde for 10 min at room temperature. Cross-linking was terminated with 2.5 M glycine for 5 min. Cell pellets were recovered by centrifugation at 1350 × *g* for 5 min at 4 °C. Cell pellets were recovered and chromatin was isolated and sonicated for 17 cycles of 30 s with an output of 30 W, using an automated sonicator (3000; Misonix)[68]. Immunoprecipitation was carried out overnight at 4 °C using 5 µg of H3K4Me1 (Abcam) or 10 µg of H3K27Ac (Abcam) or RNA Pol II N20 (Santa Cruz) antibodies previously bound to Dynabeads protein G (Thermo Fisher Scientific). DNA-protein complexes were eluted in 200 µl of elution buffer (50 mM Tris pH 8, 1 mM EDTA, 1% SDS) by incubating at 65 °C with brief agitation every 2–3 min. Cross-links were reverted both in the immunoprecipitated samples and the whole-cell extract, separated from the sonicated material before immunoprecipitation, by incubating overnight at 65 °C. DNA was then purified by phenol-chloroform extraction. Libraries were prepared using NEXTflex Rapid DNA-Seq Kit (Bioo Scientific). Four replicates per sample were sequenced on an Illumina HiSeq 2500 platform. Each ChIP-seq experiment was performed once with four technical replicates per sample.

**Cell cycle profiling.** Human primary keratinocytes and FaDu cells were collected with trypsin, fixed in ice-cold 70% ethanol overnight. Next day they were centrifuged, resuspended in PBS with DAPI, and analyzed with a LSRFortessa cell analyzer (BD bioscience). Analysis of FACS data was performed using FlowJo software (FlowJo LLC). G1, G0, S, and G2M peaks were manually delimited on the 405 nm histogram. Statistical analysis was performed in Prism (GraphPad Softwares inc.).

**Metabolic RNA labelling and isolation**. 4SU labelling and RNA isolation of newly transcribed was performed as described in ref. [69]. Briefly, for new RNA labelling with 4SU human primary keratinocytes were cultured and transfected as described above. At the indicated time point 4SU was added to the culture media at a concentration of 583 μM for 10 min. After the indicated incubation time, cells were quickly harvested by adding TRIZOL directly to the plates, and RNA was extracted following the manufacturer protocol, followed by DNase treatment and phenol-chloroform extraction. For isolation of 4SU labelled RNA, 80 μg of RNA were used per replicate. The incorporated 4SU was biothinylated by incubating 1.5 h at room temperature with EZ-link biotin-HPDP (Pierce) in biothinylation buffer (10 mM Tris Ph 7.4, 1 mM EDTA). Biothinylated RNA was then purified by phenol-chloroform extraction and loaded onto micro-MACS streptavidin columns (Miltenyi biotech). After three washes with warm (65 °C) and three with room temperature washing buffer (100 mM Tris pH 7.4, 10 mM EDTA, 1 M NaCl, 0.1% Tween 20), 4SU labelled RNA was then eluted directly into RLT buffer (Qiagen) by adding twice 100 μl of 100 mM DTT (Sigma Aldrich) and processed using the kit RNeasy MinElute (Qiagen), following manufacturer instructions.

**MNase protection assay**. Cells were collected with trypsin, quantified, and centrifuged. The pellets were then lysed in 1 ml of NP-40 lysis buffer (10 mM Tris-HCl pH 7.4, 10 mM NaCl, 3 mM $MgCl_2$, 0.5% NP-40, 0.15 mM spermine, 0.5 mM spermidine) per $10^6$ cells on ice for 5 min. After centrifugation, the pellets were then resuspended in MNase digestion buffer (10 mM Tris-HCl pH 7.4, 15 mM NaCl, 60 mM KCl, 0.15 mM spermine, 0.5 mM spermidine), centrifuged, and resuspended again in the same buffer supplemented with 1 mM $CaCl_2$ and 120 units of MNase (New England Biolabs). Samples were incubated 30 min at 37 °C. Digestion was stopped by adding an equal volume of one part of STOP buffer (100 mM EDTA, 10 mM EGTA pH 7.5) and three parts of digestion buffer. Nucleosomes were removed by proteinase K digestion. Monosomes were gel extracted using the Gel extraction kit (Qiagen) and DNA was diluted to 5 ng/μl prior to quantitative PCR. The signal of the PCR was normalized to total sonicated DNA (Input) prepared as described in the ChIP section. Four replicates of each sample were processed, and the experiment was performed once.

**Statistical analysis**. Quantified data are expressed as mean ± SD, unless otherwise stated in the figure legends. Statistical significance between samples was assessed using unpaired two-tailed Student's t-tests with Welch's correction unless otherwise stated in the figure legends. For cell cycle profiles two-way ANOVA with multiple comparisons was used. Quantitative data were analyzed using Excel and/or Prism software, with the exception for sequencing data.

**RNA library generation and sequencing**. For RNA sequencing, RNA was extracted from cells using TRIZOL (Thermo Fisher Scientific), following manufacturer instructions. After isolation, RNA was treated with turbo DNase (Thermo Fisher Scientific) for 30 min at 37 °C. DNase was subsequently removed by phenol-chloroform extraction. For total RNA sequencing at 48 h ribosomal RNA was depleted using the Ribo-Zero kit (Cambio), following manufacturer instruction. Ribo-depleted RNA was then processed with NEXTflex directional RNA-seq Kit (dUTP-Based) v2 (bio Scientific), following manufacturer instructions. Four replicates per sample were sequenced on an Illumina HiSeq 2500 platform. 4SU RNA sequencing and total RNA sequencing at 24 h libraries were prepared using the SMARTer Stranded Total RNA-Seq Kit, Pico Input Mammalian kit (Takara biotech), following manufacturer instructions. Four replicates per sample were sequenced on an Illumina HiSeq 4000 platform. Each experiment was performed once with four technical replicates per sample.

**RNA-seq analysis**. Paired-end RNA-seq reads (for 4SU and total RNA) were quality-trimmed using Trim Galore!, and mapped to the human reference genome (GRCh37/hg19) using TopHat2 with the parameters "--max-multihits 1 --read-mismatch 2 --b2-sensitive". Ensembl (release 74) gene models were used to guide alignments with the "-GTF" option. Read counts for genes or first exons were obtained using featureCounts with the parameters "-p -s 1 -O --minOverlap 10 -B -C". Read counts were normalized, and the statistical significance of differential expression between RN7SK and WT was assessed using the R Bioconductor DESeq2 package. To also identify a set of significantly unchanged genes for the Pol II ChIP-seq normalization, we did a separate analysis under the null hypothesis that $log_2FC > 0.3$. Gene counts - normalized by DESeq2 size factors - were subsequently normalized by their effective transcript length/1000, and $log_2$-transformed. Effective transcript lengths were obtained with featureCounts. Gene ontologies were calculated using the EnrichR online tool at http://amp.pharm.mssm.edu/Enrichr, selecting GO biological process 2015 or 2018, g:Profiler at https://biit.cs.ut.ee/gprofiler/gost or GOrilla at http://cbl-gorilla.cs.technion.ac.il using all transcribed genes as a background.

**ChIP-seq analysis**. Single-end ChIP-seq reads for PolII, H3K27ac, H3K4me1 ChIP, and WCE (whole-cell extract) were quality-trimmed using Trim Galore!, and reads were mapped to the human reference genome (GRCh37/hg19) using bowtie with the parameters "-m 1 -v 2" to generate unique sequence alignments. Potential PCR duplicates were removed with MACS2 'filterdup'. Narrow peaks for H3K27ac versus WCE control were called by using MACS2 'callpeak' with parameters "-B

--call-summits -q 0.05"; broad peaks were called for H3K4me1 versus WCE control with parameters "--broad --broad-cutoff 0.05". The overlapping peaks for the replicates were merged using bedops (option -m).

**Bioinformatics analyses**. Genome browser shots and metagene profiles were prepared using DeepTools. For this, the aligned reads were converted to bigWig using bamCoverage. The Pol II ChIP-seq data was processed with the following parameters: "--binsize 1 --ignoreForNormalization chrM --extendReads 150 --centerReads --normalizeUsingRPKM". Read coverage per scaled gene was calculated using computeMatrix with the parameters "scale-region --binSize 10 -b 1000 -a 500 --regionBodyLength 500 --unscaled5prime 1000 --unscaled3prime 500". The regions were defined by the start and end coordinates of all protein-coding genes (Gencode v19). To remove differences caused by variation in ChIP efficiency, we normalized each library using a set of 199 high-expressed genes that were unchanged in RN7SK knock-down in total RNA-seq (details in the RNA-seq analysis section). Specifically, we first subtracted the mean baseline signal across the gene body of those genes (excluding the regions closest to the TSS and TTS) and then divided the resulting signal by the mean across the TSS ± 1 kb. The 4SU-seq data was processed using "--binsize 1 --ignoreForNormalization chrM --normalizeUsingRPKM" for all reads or for the forward and reverse strand separately. Read coverage across all TSSs was calculated using computeMatrix with the parameters "reference-point --binSize 1 -b 500 -a 500" and a list of TSS coordinates. For this, all exons in protein-coding genes ($n = 1,071,216$) were annotated as first, internal and/or last. A set of unique exons that were exclusively first ($n = 126,731$) were identified and used.

To calculate Pol II travelling ratios (TR), aligned and duplicate-filtered PolII ChIP-seq reads were extended by 150 nt. The PolII travelling ratio (TR) was calculated as described[12], as the fraction of the PolII ChIP-seq read counts in the (1) promoter-proximal region from 0 bp to +250 bp of the annotated transcriptional start site versus the PolII ChIP-seq read counts in the (2) transcribed region from +250 bp to the annotated transcriptional end site.

To generate enhancer heatmaps, cell-type-specific enhancers were defined as H3K4me1 peaks, which did not overlap a promoter region (from −900 to +100 bp of the annotated transcriptional start site) of an Ensembl transcript. Binding profiles and heatmaps were calculated around the centre of the H3K4me1 peaks +/−3 kb using deepTools.

To model RNA synthesis, processing and degradation we used the INSPEcT tool. Exon and intron counts were derived for each nascent and total RNA-seq replicate using the quantifyExpressionsFromBAM module in strand-specific mode. RN7SK knock-down and control were treated as two separate steady-state conditions and the labelling time was set to 10 min. Genes with significant (padj < 0.05) differences in synthesis, processing, and degradation were extracted using the compareSteady module.

The ATAC-seq samples from human keratinocytes[36] were downloaded from the sequencing read archive (SRX971579 and SRX971580). Adapter (Nextera) and quality trimming were done with Trim galore! and the samples were aligned to the human genome (hg19) using bowtie (-y -m 1 -S -X 2000). PCR duplicates were removed with Picard MarkDuplicates. Binding profiles and heatmaps around the TSS +/−3 kb were calculated using DeepTools. The alignment files were converted to bigWig using bamCoverage with base-pair resolution and RPKM normalization. The read coverage per scaled gene was calculated using computeMatrix with the parameters "reference-point -b 3000 -a 3000". The regions were defined by the TSS of all protein-coding genes (hg19; Gencode v19). The up- and downregulated subsets were defined by padj < 0.05 and abs($log_2FC$) > 0.3. The heatmap was generated using plotHeatmap and the profile using custom R scripts. The null distribution profile was defined as the 99.9% confidence interval of 10,000 randomly selected gene subsets of similar size to the up- and downregulated sets. The two ATAC-seq replicates showed a very strong similarity and were therefore combined in the final figure.

To calculate bidirectional transcription in promoters we used the FANTOM5 database containing CAGE-seq data for transcriptional initiation sites across different tissues and cell lines (http://fantom.gsc.riken.jp/5/tet; "Expression (RLE normalized) of robust phase 1 and 2 CAGE peaks for human samples with annotation (hg19)"). Three replicates for human keratinocytes were used to identify transcriptional initiation across the genome: http://fantom.gsc.riken.jp/5/sstar/FF:11349-117G8, http://fantom.gsc.riken.jp/5/sstar/FF:11421-118F8, http://fantom.gsc.riken.jp/5/sstar/FF:11272-116H3.

4SU RNA seq up- and downregulated genes (FC 0.3) in RN7SK-depleted cells were screened for CAGE peaks on the opposite strand within 100 nucleotides from the annotated TSS.

The intron/exon ratios were calculated based on Gencode v19 annotations. Introns were defined for each gene by subtracting exonic regions from the full transcript and adding the resulting regions as introns to the gtf file. Raw intron and exon read counts were normalized using DESeq2-derived size factors based only on the exonic counts. Total exon and intron lengths per gene were obtained from the featureCounts output. The intron/exon ratio was calculated per gene as mean intron coverage divided by mean exon coverage, where mean coverage is the number of reads mapping to a feature divided by the feature-length. Only high-expressed genes with mean counts per million above 20 were included. Genes with extreme intron/exon ratios resulting from either no exon counts or (nearly) no intron counts were replaced by 1 or 0.001, respectively.

Differential splicing usage was computed with rMATS (v4.1.0) using "--variable-read-length --novelSS --allow-clipping". Splicing events with FDR < 0.05 and absolute difference inclusion level > 0.1 were reported.

**Reporting summary**. Further information on research design is available in the Nature Research Reporting Summary linked to this article.

## Data availability

The data supporting the findings of this study are available from the corresponding authors upon reasonable request. The next-generation sequencing (NGS) data generated in this study have been deposited in the GEO database under accession code GSE101217. ATAC-seq data from human keratinocytes are available through GEO (GSE67382; sample GSM1645708 and GSM1645709). CAGE-seq data from human keratinocytes are available through the FANTOM5 Table Extraction Tool (sample FF:11349-117G8, FF:11421-118F8, and FF:11272-116H3; http://fantom.gsc.riken.jp/5/tet). The underlying raw data generated in this study and the subsequent statistical analyses for each experiment are provided in the Supplementary Information/Source Data file. Source data are provided with this paper.

## Code availability

Custom codes used in the paper are available on GitHub (https://github.com/susbo/Bandiera-et-al-2021-scripts).
Trim Galore! https://www.bioinformatics.babraham.ac.uk/projects/trim_galore
TopHat2 https://ccb.jhu.edu/software/tophat
featureCounts http://bioinf.wehi.edu.au/featureCounts
DESeq2 https://bioconductor.org/packages/release/bioc/html/DESeq2.html
Bowtie http://bowtie-bio.sourceforge.net
MACS2 https://github.com/taoliu/MACS
rMATs https://github.com/Xinglab/rmats-turbo
bedops https://bedops.readthedocs.io
deepTools https://deeptools.github.io

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

## Acknowledgements
We thank everybody who provided us with reagents. In particular, we thank Arndt G. Benecke and Sebastian Eilebrecht for providing the *RN7SK* plasmid. We gratefully acknowledge the support of all the Wellcome MRC Cambridge Stem Cell Institute core facility managers, in particular Maike Paramor, Bill Mansfield, Gemini Chu, Peter Humphreys, and Andy Riddell. We also thank John Brown for his help obtaining tissue samples. This work was funded by Cancer Research UK (CR-UK), Worldwide Cancer Research, the Medical Research Council (MRC), the European Research Council (ERC), and the Helmholtz Association. For this research, Michaela Frye's laboratory was also supported by a core support grant from the Wellcome Trust and MRC to the Wellcome Trust-Medical Research Cambridge Stem Cell Institute. Some parts of figure panels are created with BioRender.com

## Author contributions
R.B., M.F., and S.B. designed experiments, analyzed data, and wrote the paper. R.W. performed experiments and analyzed data. T.B.-B. and S.D. performed computational analyses. C.D. performed and supervised analyses.

## Funding

## Competing interests
The authors declare no competing interests.
