## [Peer Review File · Nature Communications]

Rn7sk small nuclear RNA controls bidirectional transcription of highly expressed gene pairs in skinReviewers' comments:

Reviewer #1 (Remarks to the Author):

221973_1 Study: RNA polymerase II pausing orchestrates gene transcription with splicing

Bandiera et al. have used ChIP-seq, RNA-seq and 4sU-seq coupled with knockdown (KD) of the long noncoding RNA Rn7sk to investigate the role of Rn7sk in keratinocytes differentiation. They have noted that the presence of Rn7sk is involved in reprogramming by modulating splicing efficiency. It is an important challenge to address the precise mechanisms by which promoter proximal pausing of RNA polymerase II (Pol II) influences pre-mature RNA splicing during development. However, the data presented in this study do not allow to unambiguously draw conclusions. The methodology of the experiments and validation have several major and minor issues, listed in detail below. Unless these are addressed comprehensively we cannot recommend this manuscript for publication.

Major concerns:

1. The main conclusions regarding altered promoter-proximal pausing cannot be drawn by using pausing indices alone. Pol II occupancy (measured here by ChIP) at the 5' end of genes depends on promoter-proximal pausing but it does not directly relate to the kinetics of pausing (Ehrensberger et al. Cell 2013, PMID: 23953103). This is because the occupancy signal at a given time depends on the number of polymerases and their speed. The use of the pausing index does not allow to draw kinetic conclusions i.e. changes of Pol II pause duration, or altered pause release. The reduced Pol II occupancy at the 5' end could be explained equally well by a lower initiation frequency (Ehrensberger et al. Cell 2013). A decrease in initiation frequency would result in less RNA output per time. It was shown that promoter-proximal pausing and initiation are linked (Shao and Zeitlinger Nature Genet 2017, PMID: 28504701; Gressel et al. eLife 2017, PMID: 28994650). In addition, the increased signal in the gene body as well as around the TTS after Rn7sk KD (e.g. Fig. 2 B and Fig. 3 G-H) could indicate slower Pol II speed in the gene body (compared to Ctrl). In summary, using only occupancy data the authors cannot conclude how/if pause release rates are changed after KD. This is only possible when the transcriptional output per time is considered (Ehrensberger et al. Cell 2013; Gressel et al. eLife 2017). The latter is also evident in the data generated by the authors (Fig. 6): while Pol II occupancy (measured by ChIP) changes in a similar way for up- and downregulated genes (less 5' occupancy, more gene body/3' occupancy upon KD), the transcription activity (measured by 4sU-seq) differs. How do the authors explain this observation?

2. How can the authors distinguish effects of promoter-proximal pausing (partially addressed by the authors using pausing indices) versus elongation rate (not addressed, requires a measure of elongation rate, e.g. 4sUDRB-seq) on splicing? Both processes were shown to modulate splicing efficiency: it was shown recently that depletion of the nuclear exon junction complex (EJC) or the splicing regulator SRSF2 results in altered promoter-proximal Pol II pausing, and changes in exon usage (Akhtar et al. Nature Comm 2019, PMID: 30705266; Ji et al. Cell 2013, ref 13). On the other hand, it was shown by several labs that changes in elongation rate lead to alternative splicing (de la Mata et al. Mol Cell 2003; Munoz et al. Cell 2009; Close et al. Nature 2012; Dujardin et al. Mol Cell 2014; Godoy et al. Mol Cell 2019).

3. It was recently reported that Rn7sk KD inhibits neuronal differentiation (Bazi et al. 2018, PMID: 29091296). How does this compare to the observation in this study that Rn7sk KD induces differentiation? It would be interesting and an important control to compare Rn7sk RNA levels along the normal keratinocyte differentiation (Ctrl samples on day 0, 2, 3). In general, the authors should briefly summarize in the introduction which observations were made so far using Rn7sk overexpression or KD (see also Castelo-Branco et al. Genome Biol 2013, PMID: 24044525).

Specific comments:

1. It is a significant challenge to accurately annotate sequences as "active" enhancers. Enhancer definitions vary and recent evidence showed that chromatin marks may be misleading in the classification of enhancer vs. promoter (Henriques et al. *Genes Dev* 2018; Dao et al. *Nature Genet* 2017; Diao et al. *Nature Meth* 2017; reviewed in Halfon *Trends Genet* 2019). Here only H3K4me1 was used to classify enhancers (Methods, p. 26). Since H3K27ac data are available, the authors could include these to call putative enhancers. In summary, the authors cannot derive enhancer activity from chromatin marks, and should refer to these as putative enhancers.

2. ChIP-seq normalization: it is puzzling that up- and downregulated genes show the same occupancy change (see also above, major concern no. 1). This could be due to a normalization artefact. It would be recommended to re-analyse the data sets using normalization on "unchanged" genes as described e.g. by Mahat et al. *Mol Cell* 2016, PMID: 27052732.

3. It is unclear how the size factor was calculated for PlaB samples. The description in the Methods sections sounds like DESeq2 normalization. However, to track changes in splicing the authors should calculate size factors only on the exons (-t exon), and apply the resulting size factor to normalize the intron levels. Subsequently DE analysis can be performed.

4. Abstract, line 4-5: transcriptional bursting can be influenced by many regulatory steps of transcription (see e.g. Nicolas et al. 2017, PMID 28573295). Furthermore, to describe bursting, several parameters (i.e. burst duration, size and frequency) are critical. Thus, we wonder if there is sufficient evidence that pausing is a key step in the regulation of bursting? Specifically, what parameters may be controlled by pausing (compared to the initiation frequency)?

5. Introduction, line 12-15: biochemical evidence (Vos et al. 2018, PMID: 30135580 and 30135578; summarized in Adelman 2018, PMID: 30143755) shows that the paused elongation complex adopts a nonproductive conformation of the RNA-DNA hybrid which is stabilized by DSIF (Spt5) and NELF. Thus, the adoption of the paused state is not due to negative regulation of P-TEFb. The release of the paused elongation complex requires that P-TEFb (Cdk9) is present (1st point of regulation by P-TEFb recruitment/delivery, cite here e.g. Baboric et al. 2001, PMID: 11545735; Rahl et al. 2010, PMID: 20434984; Takahashi et al. 2011, PMID: 21729782) and active (2nd point of regulation by the inactive 7SK snRNP complex). This is not always clearly stated throughout the text. Please rephrase esp. the abstract and discussion accordingly.

6. Introduction, line 12-15: note, that it is an open question in the field (due to lack of specific antibodies) if P-TEFb (Cdk9) phosphorylates serine 2. It was also shown to phosphorylate serine 5 as well as the Pol II linker to the CTD (Vos et al. 2018, PMID: 30135578). Please reword accordingly.

7. What does "presence of RNA Pol II pause release" mean? Promoter-proximal pause durations vary between genes, as well as between conditions - even if the pause release factors are present.

8. Experimental set-up and analysis.

- Biological replicates: it is unclear if biological or technical replicates were performed for ChIP-seq, RNA-seq and 4sU-seq experiments. Please clarify this. At least two biological replicates should be performed per condition.

- qPCR and RNA-seq data normalization: (i) did the authors test if GAPDH (RT-qPCR) is not affected by changes in Rn7sk (since pausing is a genome-wide phenomenon)? In general, it would be good if at least two housekeeping genes are compared, or synthetic spike-ins are added which allow for unbiased normalization. (ii) The use of spike-ins is also critical for the RNA-seq and 4sU-seq, especially when global transcription factors are manipulated (Loven et al. *Cell* 2012, PMID: 23101621).

- ChIP-seq data: it is not clear if replicates were merged for the analysis, and for metagene plots how

many genes were analysed (Fig. 2 B, 3 G-H). Please add this information to the figure legend. In Fig. 4 I-J it is unclear why only one out of four replicates is shown. How does the metagene analysis look for merged replicates? In general, it would be good if confidence intervals could be added to the average signal of the metagene plots to interpret Ctrl vs. KD.

- "Data not shown" (p. 5, 6): for transparency this statement should be avoided. If data are not provided the respective statements should be removed. Since the transgenic mouse lines are key to the findings of the paper, viability data should be added to the Supplementary Information (the authors claim that both did not show "any gross phenotype").
- Why are Rn7sk KD times different for each experimental set-up? It would be useful for the reader to add a supplementary table providing an overview of all the conditions used in this study.
- Reference genome: why are the authors using the hg19 annotation for the RNA-seq and ChIP-seq data sets (the updated annotation hg38 is available since 2013, and lifting older data sets for comparison is fast via <http://genome.ucsc.edu/cgi-bin/hgLiftOver>), but use hg38 for ATAC-seq and the GC content calculation?
- Results: did the authors control if Larp7 is also depleted on the protein level upon knock-down? If yes, please add the respective Western blot to Fig. S2C.
- Results (p. 9, line 14-16; Fig. S4 C-D): the rescue experiment is a critical control. The statement that repression of cell cycle regulators was prevented appears to be not significant for Cdk1. Please rephrase the sentence accordingly, and test significance as done in A-B.

9. Missing information.

- Several figures are missing information which statistical test was used and which p-value corresponds to asterisk(s) to the figure legend. Please add the missing information e.g. for Figures S1, S4, S5, S6.
- Methods, all -seq experiments: add information how many cycles were sequenced, and if single or paired-end mode was used.
- Several figures are missing information how many genes are depicted in the metagene profiles (n=?). E.g. Fig. 2 B: please add the number of protein-coding genes (n=?) depicted in the metagene plot. The same issues for figures 3, 4, 6.
- Fig. 2 H-J: it is confusing that the FC (H), ? (I), or log2 FC (J) are shown. Please add the missing title of the y-axis in (I).
- Fig. S4 A: which siRNA was used?
- Fig. 5 B: please add genomic coordinates and information which genome browser was used.
- Methods: were duplicates removed for RNA-seq and 4sU-seq? Please add this information to the methods.
- Fig. S4 B: please add which cell line was used to compare the 3 siRNAs.

10. Missing references.

- Introduction, line 2-4: please add Gariglio et al. 1981, PMID: 6269056 - they were the first describing the accumulation of Pol II at the beta-globin locus.
- Introduction, line 5 and 12: please add Core and Adelman 2019, PMID: 31123063 - it's the most recent review on promoter-proximal events.
- Introduction, line 17-19: please add Shao and Zeitlinger Nature Genet 2017, PMID: 28504701 and Gressel et al. 2017, PMID: 28994650 - they were the first describing that pausing can also limit initiation, and thus, regulate transcriptional activity.
- Methods, p. 21, line 5: add reference for "as previously described".
- Discussion, p. 15, line 5: the impact of promoter-proximal pausing on bursts was shown by Ehrensberger et al. Cell 2013 as well as Shao and Zeitlinger Nature Genet 2017. Please add these.

11. Other comments.

- Spelling: results, line 18: "ChIP"
- Fig. 1 + 3: what statistical test is referred to by "multiple t-test"?

Reviewer #2 (Remarks to the Author):

Bandiera et al. address the important question how pausing of RNA polymerase II (Pol II), a key regulatory step in RNA synthesis, affects gene expression and adult tissue homeostasis. The authors present several remarkable observations that will largely advance our current understanding how RNA synthesis regulation is mechanistically tied to cellular differentiation.

Most interestingly, the observation that the forced release of transcription by depleting the noncoding RNA Rn7sk results in repressed RNA levels is unexpected and could be convincingly linked to splicing defects and co-transcriptional RNA degradation. Further, it is demonstrated that Rn7sk, a crucial component of the ribonucleoprotein complex coordinating Pol II pausing, is specifically required for the correct expression of a distinct set of genes that are marked by highly accessible promoters, lower guanine/cytosine content, shorter introns and weaker 3' splice sites. In particular, the Rn7sk depletion results in repression of cell cycle regulators and promotes cellular differentiation. Although the data are exciting, novel and appear robust as the authors explore two experimental systems: genetic manipulation of mouse epidermis *in vivo* and human primary keratinocytes *in vitro*, some critical points need to be addressed.

1. The authors demonstrate that Rn7sk depletion leads to reduced expression of integrin alpha 6 (figure 2 H). In this context it needs to be investigated if some of the observed phenotypes, including colony forming potential, epidermal reconstitution (ex vivo assay on de-epidermalised dermis) as well as the differentiation and changes in cellularity seen in Rn7sk cKO mice is a direct result of integrin regulation (rather than a specific response to Pol II pausing).
2. The direct functional link between cell cycle gene regulation and induction of terminal differentiation in Rn7sk-depleted keratinocytes needs to be further clarified. Are cell cycle regulation and differentiation independently controlled by Rn7sk activity? Can the authors exclude that cellular stress response mechanisms lead to the phenotype, e.g. change in cellularity *in vivo* and *in vitro*?
3. The recovery of the phenotype and wound-like response, despite complete absence of Rn7sk seen *in vivo* is interesting. However, it needs to be shown that recovery of the epidermis indeed occurs in Rn7sk-depleted tissue.

Reviewer #3 (Remarks to the Author):

This paper describes the effect of knocking out the 7SK RNA gene in transgenic mice epidermis (by keratinocyte selective KO) or depleting 7SK RNA by siRNA in an epidermal cell line, which normally acts to restrict Pol II elongation by sequestering P-TEFb. In both systems a clear cell growth defect is detected which in the cell line study is attributed to a cell cycle defect. Thus, 7SK depletion causes a selective down regulation in the expression of genes involved in cell cycle progression. Overall, I felt that the data presented in Fig 1-3 looks reliable and convincing (as far as this non-expert reviewer in epidermis differentiation is concerned). The data presented in Fig 3 clearly shows that while as expected 7SK depletion reduces TSS associated Pol II pausing, surprisingly many genes are down regulated in expression. To seek an explanation for this apparent data contradiction, initially chromatin analysis was tested to see if these cell cycle associated genes displayed repressed chromatin features. However as shown in Fig 4 apparently 7SK depletion has no significant detectable effects on chromatin structure over relevant promoters or enhancers. Again, I found these data very credible.

Specific concerns:

- 1) Figure 5 shows by isolating pulse labelled nascent RNA (4SU) that 7SK depletion does reduce nascent transcription levels. I think these data would benefit from more gene screen shots clearly showing this reduction in nascent transcription. The TP63 gene data is hard to appreciate (Fig 5B).

Really it would be good to have back up data using GRO-seq or mNET-seq here.

2) I have serious misgivings about some of transcriptomic analysis related to splicing aimed at understanding the mechanism behind the 7SK kd phenotypes. Principally I think 4SU labelling is problematic for looking at splicing as it is possible that this analogue incorporation into RNA may impair splicing which relies on splice site base pairing to snRNAs. Really these data need repeating using other nascent transcription analysis methods that will give unbiased splicing efficiency measures. Possibly a comparison of chromatin vs nucleoplasm RNA looking for intron retention (+/- 7SK depletion). Alternatively, mNET-seq using the splicing specific Pol II CTD S5P selection would be a good approach (see Nojima et al. Mol Cell 72,369 2018).

3) I am unsure if the bioinformatic correlation of low GC content and weaker 3' splice sites for 7SK depletion repressed genes is a very meaningful correlation.

4) I don't understand why PlaB treatment which will clearly block all splicing has an antagonistic effect on 7SK depletion induced splicing defects. Surely the combination of 7SK depletion followed by PlaB treatment should cause additive effects on splicing inhibition, not rescue effects? Is this again related to potential problems with using 4SU labelling to study splicing efficiency. Gene specific screen shots are needed here to present these data in a more convincing manner.

Overall, I feel that while the first parts of this paper look compelling and interesting I don't consider the splicing connection is convincingly made. More work is needed as suggested above to either confirm or rule out a splicing connection.

NCOMMS-19-11141432

In response to referees' comments, we have made the following changes to our manuscript.

Reviewers' comments:**Reviewer #1:**

221973_1 Study: RNA polymerase II pausing orchestrates gene transcription with splicing

Bandiera et al. have used ChIP-seq, RNA-seq and 4sU-seq coupled with knockdown (KD) of the long noncoding RNA Rn7sk to investigate the role of Rn7sk in keratinocytes differentiation. They have noted that the presence of Rn7sk is involved in reprogramming by modulating splicing efficiency. It is an important challenge to address the precise mechanisms by which promoter proximal pausing of RNA polymerase II (Pol II) influences pre-mature RNA splicing during development. However, the data presented in this study do not allow to unambiguously draw conclusions. The methodology of the experiments and validation have several major and minor issues, listed in detail below. Unless these are addressed comprehensively we cannot recommend this manuscript for publication.

We thank the referees for the in-depths review of our manuscript and the constructive criticism helping us to improve our manuscript. We have now extensively re-analysed our data and provide novel mechanistic insights how *Rn7sk* directly regulates gene-specific transcription initiation (see detailed responses to major concerns below).

Major concerns:

1. The main conclusions regarding altered promoter-proximal pausing cannot be drawn by using pausing indices alone. Pol II occupancy (measured here by ChIP) at the 5' end of genes depends on promoter-proximal pausing but it does not directly relate to the kinetics of pausing (Ehrensberger et al. Cell 2013, PMID: 23953103). This is because the occupancy signal at a given time depends on the number of polymerases and their speed. The use of the pausing index does not allow to draw kinetic conclusions i.e. changes of Pol II pause duration, or altered pause release. The reduced Pol II occupancy at the 5' end could be explained equally well by a lower initiation frequency (Ehrensberger et al. Cell 2013). A decrease in initiation frequency would result in less RNA output per time. It was shown that promoter-proximal pausing and initiation are linked (Shao and Zeitlinger Nature Genet 2017, PMID: 28504701; Gressel et al. eLife 2017, PMID: 28994650). In addition, the increased signal in the gene body as well as around the TTS after *Rn7sk* KD (e.g. Fig. 2 B and Fig. 3 G-H) could indicate slower Pol II speed in the gene body (compared to Ctrl). In summary, using only occupancy data the authors cannot conclude how/if pause release rates are changed after KD. This is only possible when the transcriptional output per time is considered (Ehrensberger et al. Cell 2013; Gressel et al.

eLife 2017). The latter is also evident in the data generated by the authors (Fig. 6): while Pol II occupancy (measured by ChIP) changes in a similar way for up- and downregulated genes (less 5' occupancy, more gene body/3' occupancy upon KD), the transcription activity (measured by 4sU-seq) differs. How do the authors explain this observation?

We agree with the referee that our conclusion that repression of *Rn7sk* altered promoter-proximal pausing cannot be drawn from pausing indices alone. Therefore, we also quantified Pol II phosphorylation changes at serine 2 and 5 by Western blotting (Figure 3C). We find the same level of serine 5 phosphorylation, indicating that transcription initiation is unchanged upon depletion of *Rn7sk*. A two-fold increase in serine 2 phosphorylation indicated enhanced productive elongation. Reduced levels of Pol II occupancy cannot explain these results because the overall Pol II levels are unchanged (Figure 3C).

A lower Pol II initiation frequency should increase the pausing duration (Gressel et al., 2019; Gressel et al., 2017; Shao and Zeitlinger, 2017). We consistently find reduced pausing indices in *Rn7sk*-depleted cells (Figure 3B; Figure S3C, D).

In addition, lower initiation frequencies without pausing are likely to cause closed or repressive chromatin architectures (Core and Adelman, 2019). However, we find no evidence for changes in chromatin architecture in *Rn7sk*-depleted cells (Figure 5A-J; Figure S5A, B). Therefore, we disagree with the referee that lower occupancy of Pol II alone could explain our findings.

We agree with the referee that an increased signal in the gene body as well as around the TTS after *Rn7sk* knockdown could indicate slower Pol II speed in the gene body. However, we do not find evidence for significant slower global elongation (Figure 3A). We would also expect to see a reduction of serine 2 phosphorylation, which was not the case (Figure 3C).

In summary, we entirely agree with the referees that it is challenging to explain our data with the current knowledge that 7SK ribonucleoprotein complex sequesters P-TEFb. Please note, that previous studies identified an inhibitory role on P-TEFb in response to ultraviolet radiation, yet a recent study also finds little changes in global transcription upon *Rn7sk*-depletion in the absence of stress (Nguyen et al., 2001; Studniarek et al., 2021; Yang et al., 2001).

As requested by the referees, we now provide further analyses demonstrating a functional role of the 7SK snRNP complex in transcription initiation specifically of highly transcribed bi-directional gene pairs. We have now also re-written our manuscript to better highlight the novel functional roles of *Rn7sk* instead of focusing entirely on Pol II pausing.

2. How can the authors distinguish effects of promoter-proximal pausing (partially addressed by the authors using pausing indices) versus elongation rate (not addressed, requires a measure of elongation rate, e.g. 4sUDRB-seq) on splicing? Both processes were shown to modulate splicing efficiency: it was shown recently that depletion of the nuclear exon junction complex (EJC) or the splicing regulator SRSF2 results in altered promoter-proximal Pol II pausing, and changes in exon usage (Akhtar et al. Nature Comm 2019, PMID: 30705266; Ji

et al. Cell 2013, ref 13). On the other hand, it was shown by several labs that changes in elongation rate lead to alternative splicing (de la Mata et al. Mol Cell 2003; Munoz et al. Cell 2009; Close et al. Nature 2012; Dujardin et al. Mol Cell 2014; Godoy et al. Mol Cell 2019).

The referee is correct, both promoter proximal pausing and changes in transcription elongation rate could contribute to our findings. 7SK has been described to also prevent transcription downstream of polyadenylation sites at several active genes (Castelo-Branco et al., 2013), and we detected a significant enrichment of sense genes downstream of *Rn7sk*-dependent genes (*Figure 4C*). However, because this effect was less pronounced we focused on the role of *Rn7sk* in bidirectional transcription.

3. *It was recently reported that Rn7sk KD inhibits neuronal differentiation (Bazi et al. 2018, PMID: 29091296). How does this compare to the observation in this study that Rn7sk KD induces differentiation? It would be interesting and an important control to compare Rn7sk RNA levels along the normal keratinocyte differentiation (Ctrl samples on day 0, 2, 3). In general, the authors should briefly summarize in the introduction which observations were made so far using Rn7sk overexpression or KD (see also Castelo-Branco et al. Genome Biol 2013, PMID: 24044525).*

Bazi and colleagues depleted 7SK during an embryonic stem cell differentiation assay towards neurons. The read-out for reduced differentiation was reduction of the neural differentiation markers Map2 and Nestin (Bazi et al., 2018). The reason for lower expression of the markers in the cultured cells was not investigated. Thus, a reduction of proliferation at the neural progenitor stage may have contributed to their finding.

We show in *Figure 2E* that RNA expression levels of *Rn7sk* does not significantly change during epidermal differentiation (Ctr siRNA).

Previous *Rn7sk* over-expression and depletion studies are now summarized in the introduction.

Specific comments:

1. *It is a significant challenge to accurately annotate sequences as “active” enhancers. Enhancer definitions vary and recent evidence showed that chromatin marks may be misleading in the classification of enhancer vs. promoter (Henriques et al. Genes Dev 2018; Dao et al. Nature Genet 2017; Diao et al. Nature Meth 2017; reviewed in Halfon Trends Genet 2019). Here only H3K4me1 was used to classify enhancers (Methods, p. 26). Since H3K27ac data are available, the authors could include these to call putative enhancers. In summary, the authors cannot derive enhancer activity from chromatin marks, and should refer to these as putative enhancers.*

Cell type-specific enhancers were defined as H3K4me1 peaks, which did not overlap a promoter region (from -900 bp to +100 bp of the annotated transcriptional start site) of an Ensembl transcript (see Methods). At these sites, we find no differences in H3K4me1 or H3K27ac peaks (*Figure 5I, J*). As requested by the referee, we now refer to these sites as putative enhancers (not as active enhancers) in the text.

Our initial expectation from the literature was to find a small set of enhancers that changed in the absence of 7SK (Flynn et al., 2016). Therefore, we first considered peaks with H3K4me1 or/and H3K27Ac to identify poised enhancers (Creyghton et al., 2010). Since we did not find differences in either, we defined them by both H3K27ac and H3K4me1 marks (*Figure 5E-J; Figure S5A, B*).

2. ChIP-seq normalization: it is puzzling that up- and downregulated genes show the same occupancy change (see also above, major concern no. 1). This could be due to a normalization artefact. It would be recommended to re-analyse the data sets using normalization on “unchanged” genes as described e.g. by Mahat et al. Mol Cell 2016, PMID: 27052732.

As requested by the referee, we re-analysed our data using common ‘unchanged’ genes in all datasets (n=199) and confirm a significant genome-wide reduction in RNA Pol II occupancy at the TSS but not at the gene body (*Figure 3A*).

To address the referees’ concern how the same Pol II occupancy change can have opposite effects on RNA synthesis, we closer inspected the common features of the up- and down-regulated nascent transcripts in *Rn7sk*-depleted cells. We now show that the top up-regulated nascent RNAs often contained several alternative transcriptional start sites (*Figure 3I; upper panel; Figure S3H, J*). This could have an additive effect and lead to an accumulation of transcript reads over the gene body, even though Pol II occupancy at individual start sites is lower.

The top down-regulated genes were commonly bi-directionally transcribed (*Figure 3I; lower panel; Figure 3K, lower panel; Figure S3H, J*). Simultaneous transcriptional initiation on opposite strands is likely to require more complex regulatory processes. We now show that *Rn7sk* is indeed specifically required for efficient bi-directional transcription of highly expressed gene pairs (*Figure 4; Figure S4*). Since this finding is novel and explains the gene-specific effects on cell cycle genes for instance, we have now focused on the function of *Rn7sk* in bi-directional transcription and changed the text and title accordingly.

3. It is unclear how the size factor was calculated for PlaB samples. The description in the Methods sections sounds like DESeq2 normalization. However, to track changes in splicing the authors should calculate size factors only on the exons (-t exon), and apply the resulting size factor to normalize the intron levels. Subsequently DE analysis can be performed.

We agree with the referee and performed the normalization as the referee describes it whenever possible. We only used a slightly modified approach for some of the PlaB analyses due to the global changes in overall mRNA levels following PlaB treatment. We have now excluded the PlaB experiment and therefore no longer needed the alternative approach. The manuscript has been updated with the following sentence “Raw intron and exon read counts were normalized using DESeq2-derived size factors based only on the exonic counts.” to clarify how normalization was done.

4. Abstract, line 4-5: transcriptional bursting can be influenced by many regulatory steps of transcription (see e.g. Nicolas et al. 2017, PMID 28573295). Furthermore, to describe bursting, several parameters (i.e. burst duration, size and frequency) are critical. Thus, we wonder if there is sufficient evidence that pausing is a key step in the regulation of bursting? Specifically, what parameters may be controlled by pausing (compared to the initiation frequency)?

We agree with the referee and have omitted the sentence from the abstract.

5. Introduction, line 12-15: biochemical evidence (Vos et al. 2018, PMID: 30135580 and 30135578; summarized in Adelman 2018, PMID: 30143755) shows that the paused elongation complex adopts a nonproductive conformation of the RNA-DNA hybrid which is stabilized by DSIF (Spt5) and NELF. Thus, the adoption of the paused state is not due to negative regulation of P-TEFb. The release of the paused elongation complex requires that P-TEFb (Cdk9) is present (1st point of regulation by P-TEFb recruitment/delivery, cite here e.g. Baboric et al. 2001, PMID: 11545735; Rahl et al. 2010, PMID: 20434984; Takahashi et al. 2011, PMID: 21729782) and active (2nd point of regulation by the inactive 7SK snRNP complex). This is not always clearly stated throughout the text. Please rephrase esp. the abstract and discussion accordingly.

As requested by the referee, we have re-worded the introduction and included the citations.

6. Introduction, line 12-15: note, that it is an open question in the field (due to lack of specific antibodies) if P-TEFb (Cdk9) phosphorylates serine 2. It was also shown to phosphorylate serine 5 as well as the Pol II linker to the CTD (Vos et al. 2018, PMID: 30135578). Please reword accordingly.

As requested by the referee, we have re-worded the introduction and included the citations.

7. What does “presence of RNA Pol II pause release” mean? Promoter-proximal pause durations vary between genes, as well as between conditions - even if the pause release factors are present.

The phrase has now been omitted from the text.

8. Experimental set-up and analysis.

- Biological replicates: it is unclear if biological or technical replicates were performed for ChIP-seq, RNA-seq and 4sU-seq experiments. Please clarify this. At least two biological replicates should be performed per condition.

This information is now provided in the respective figure legends.

- qPCR and RNA-seq data normalization: (i) did the authors test if GAPDH (RT-qPCR) is not affected by changes in Rn7sk (since pausing is a genome-wide phenomenon)? In general, it would be good if at least two housekeeping genes are compared, or synthetic spike-ins are added which allow for unbiased normalization. (ii) The use of spike-ins is also critical for the RNA-seq and 4sU-seq, especially when global transcription factors are manipulated (Loven et al. Cell 2012, PMID: 23101621).

The RNA levels of GAPDH were not significantly down-regulated in of the dataset (Figure S4G). We also performed RTqPCRs using 18S rRNA as a standard. This information is now included in the Methods section.

In Fig. 4 I-J it is unclear why only one out of four replicates is shown. How does the metagene analysis look for merged replicates? In general, it would be good if confidence intervals could be added to the average signal of the metagene plots to interpret Ctrl vs. KD.

All four replicates are now provided in Figure S5A, B.

- “Data not shown” (p. 5, 6): for transparency this statement should be avoided. If data are not provided the respective statements should be removed. Since the transgenic mouse lines are key to the findings of the paper, viability data should be added to the Supplementary Information (the authors claim that both did not show “any gross phenotype”).

The data regarding the mouse viability is provided in Figure S1J. The phrase “data not shown” has been removed.

- Why are Rn7sk KD times different for each experimental set-up? It would be useful for the reader to add a supplementary table providing an overview of all the conditions used in this study.

To only study direct functions of Rn7s, we aimed to analyse the cells as soon as possible after Rn7s-knockdown. The requested information is now provided in Supplementary Table S3.

- Reference genome: why are the authors using the hg19 annotation for the RNA-seq and ChIP-seq data sets (the updated annotation hg38 is available since 2013, and lifting older data sets for comparison is fast via <http://genome.ucsc.edu/cgi->), but use hg38 for ATAC-seq and the GC content calculation?

This project has been started several years ago and therefore, the analyses have been done using hg19. Only one set of data analysis (ATAC-seq) was performed using hg38. Since there are no major differences in the annotation of protein coding genes between the two versions, we now provide all analyses in hg19.

- Results: did the authors control if *Larp7* is also depleted on the protein level upon knock-down? If yes, please add the respective Western blot to Fig. S2C.

Since the *Larp7* knockdown experiments served as an additional control to test for *Rn7sk*-specific knockdown effects, we have only measured RNA levels at the time. Since we do confirm repression of *Rn7sk* and cell cycle regulators in the absence of *Larp7* (Figure S4F; lower panel), we can assume that the reduction of RNA levels lead to reduced translation.

- Results (p. 9, line 14-16; Fig. S4 C-D): the rescue experiment is a critical control. The statement that repression of cell cycle regulators was prevented appears to be not significant for *Cdk1*. Please rephrase the sentence accordingly, and test significance as done in A-B.

We entirely agree with the referee and provided significance when applicable to all of our data. The rescue experiments in primary epidermal cells is highly challenging because these are not immortalized lines, they easily differentiate and thereby naturally repress cell cycle regulators. To address the referees' concern, we have now provided additional rescue experiments in an immortalized epidermal line (Figure 7F, G; Figure S7H). We confirm the cell cycle regulators to be directly affected by depletion of *Rn7sk* and can demonstrate that also the cell cycle alterations were directly caused by *Rn7sk* (Figure 7F, G).

9. Missing information.

- Several figures are missing information which statistical test was used and which p-value corresponds to asterisk(s) to the figure legend. Please add the missing information e.g. for Figures S1, S4, S5, S6.

All details are now provided in the source data file.

- Methods, all -seq experiments: add information how many cycles were sequenced, and if single or paired-end mode was used.

This information is available at GEO
(<http://www.ncbi.nlm.nih.gov/projects/geo/query/acc.cgi?acc=GSE101217>)

- Several figures are missing information how many genes are depicted in the metagene profiles (n=?). E.g. Fig. 2 B: please add the number of protein-coding genes (n=?) depicted in the metagene plot. The same issues for figures 3, 4, 6.

The number of genes has now been included throughout the figures.

- Fig. 2 H-J: it is confusing that the FC (H), ? (I), or log2 FC (J) are shown. Please add the missing title of the y-axis in (I).

To be consistent, we changed all graphs to log2-fold change and the missing title is now included.

- Fig. S4 A: which siRNA was used?

If not otherwise stated all experiments were done using siRNA5. This information is given in the text (line 117-118).

- Fig. 5 B: please add genomic coordinates and information which genome browser was used.

All genome browser shots were generated using the UCSC Genome Browser (<https://genome-euro.ucsc.edu>).

- Methods: were duplicates removed for RNA-seq and 4sU-seq? Please add this information to the methods.

Duplicates are commonly not removed from RNA-seq data, and we also did not do so.

- Fig. S4 B: please add which cell line was used to compare the 3 siRNAs.

Please note that the numbering of the primary keratinocyte cultures is arbitrary (alphabetical) to denote different donors. We used one line throughout the study and state in the text when we validated the experiments in different independent lines from other donors.

10. Missing references.

- Introduction, line 2-4: please add Gariglio et al. 1981, PMID: 6269056 - they were the first describing the accumulation of Pol II at the beta-globin locus.

- Introduction, line 5 and 12: please add Core and Adelman 2019, PMID: 31123063 - it's the most recent review on promoter-proximal events.

- Introduction, line 17-19: please add Shao and Zeitlinger Nature Genet 2017, PMID: 28504701 and Gressel et al. 2017, PMID: 28994650 - they were the first describing that pausing can also limit initiation, and thus, regulate transcriptional activity.

- Methods, p. 21, line 5: add reference for "as previously described".

- Discussion, p. 15, line 5: the impact of promoter-proximal pausing on bursts was shown by Ehrensberger et al. Cell 2013 as well as Shao and Zeitlinger Nature Genet 2017. Please add these.

As requested, we have included the missing citation if relevant after editing the text.

11. Other comments.

- Spelling: results, line 18: “ChIP”

The spelling mistake has been corrected.

- Fig. 1 + 3: what statistical test is referred to by “multiple t-test”?

All details are now provided in the source data file.

Reviewer #2:

Bandiera et al. address the important question how pausing of RNA polymerase II (Pol II), a key regulatory step in RNA synthesis, affects gene expression and adult tissue homeostasis. The authors present several remarkable observations that will largely advance our current understanding how RNA synthesis regulation is mechanistically tied to cellular differentiation.

Most interestingly, the observation that the forced release of transcription by depleting the noncoding RNA Rn7sk results in repressed RNA levels is unexpected and could be convincingly linked to splicing defects and co-transcriptional RNA degradation. Further, it is demonstrated that Rn7sk, a crucial component of the ribonucleoprotein complex coordinating Pol II pausing, is specifically required for the correct expression of a distinct set of genes that are marked by highly accessible promoters, lower guanosine/cytosine content, shorter introns and weaker 3' splice sites. In particular, the Rn7sk depletion results in repression of cell cycle regulators and promotes cellular differentiation. Although the data are exciting, novel and appear robust as the authors explore two experimental systems: genetic manipulation of mouse epidermis in vivo and human primary keratinocytes in vitro, some critical points need to be addressed.

We thank the referee for finding our data novel, robust, and exciting. As requested by the referee, we have now provided more evidence that *Rn7sk* directly regulates cell cycle genes leading to enhanced differentiation as a secondary effect *in vitro* and *in vivo* (see detailed response to the specific comments below).

1. *The authors demonstrate that Rn7sk depletion leads to reduced expression of integrin alpha 6 (figure 2 H). In this context it needs to be investigated if some of the observed phenotypes, including colony forming potential, epidermal reconstitution (ex vivo assay on de-epidermalised dermis) as well as the differentiation and changes in cellularity seen in Rn7sk*

cKO mice is a direct result of integrin regulation (rather than a specific response to Pol II pausing).

We agree with the referee that the down-regulation of ITGA6 might contribute to the cellular phenotype of *Rn7sk*-depletion in the *in vitro* and *in vivo* assays at later time points (i.e. longer than 48 hours) and in calcium-high conditions. However, all our mechanistic analyses in Figures 3 to 7, in which we identify the cell cycle regulators as underlying driver of the phenotype, were performed between 12 to 24 hours. At these early time points and without a differentiation stimulus, we do not measure down-regulation of ITGA6 in the 4SU or total RNA sequencing datasets (see figure panel on the right).

2. *The direct functional link between cell cycle gene regulation and induction of terminal differentiation in *Rn7sk*-depleted keratinocytes needs to be further clarified. Are cell cycle regulation and differentiation independently controlled by *Rn7sk* activity? Can the authors exclude that cellular stress response mechanisms lead to the phenotype, e.g. change in cellularity *in vivo* and *in vitro*?*

We now demonstrate that cell cycle regulators but not up-regulated differentiation markers significantly change in expression as early as 12 hours after *Rn7sk* knock-down (Figure S4C, D). Moreover, expression of cell cycle regulators and the effect on cell cycle can be rescued by re-expressing *Rn7sk* in knock-down cells (Figure S4H, I; Figure 7F, G). Thus, the referee is correct, cell cycle regulation and epidermal differentiation are independently controlled, and enhanced differentiation is highly likely a secondary effect of cell cycle regulator repression and hence *Rn7sk* knock-down.

Whether cellular stress responses contribute to the observed phenotype is an interesting question. Although we do not find genes regulating the cellular stress response to be significantly changed in our analyses (Figure 7A, B), we cannot fully exclude that the stress response contributed to the phenotype. We now show that DNA repair genes are often bi-directionally transcribed and are also enriched in *Rn7sk*-depleted cells (Figure 4H,I). Thus, a mis-regulated DNA damage and repair response might indeed contribute to the phenotype.

3. *The recovery of the phenotype and wound-like response, despite complete absence of *Rn7sk* seen *in vivo* is interesting. However, it needs to be shown that recovery of the epidermis indeed occurs in *Rn7sk*-depleted tissue.*

We provide histological sections labelling *Rn7sk* RNA in the epidermis more than 30 days after the last treatment in Figure S1I (upper panels) and in the total knockout mice in Figure S1K (upper panels).

Reviewer #3:

This paper describes the effect of knocking out the 7SK RNA gene in transgenic mice epidermis (by keratinocyte selective KO) or depleting 7SK RNA by siRNA in an epidermal cell line, which normally acts to restrict Pol II elongation by sequestering P-TEFb. In both systems a clear cell growth defect is detected which in the cell line study is attributed to a cell cycle defect. Thus, 7SK depletion causes a selective down regulation in the expression of genes involved in cell cycle progression. Overall, I felt that the data presented in Fig 1-3 looks reliable and convincing (as far as this non-expert reviewer in epidermis differentiation is concerned). The data presented in Fig 3 clearly shows that while as expected 7SK depletion reduces TSS associated Pol II pausing, surprisingly many genes are down regulated in expression. To seek an explanation for this apparent data contradiction, initially chromatin analysis was tested to see if these cell cycle associated genes displayed repressed chromatin features. However as shown in Fig 4 apparently 7SK depletion has no significant detectible effects on chromatin structure over relevant promoters or enhancers. Again, I found these data very credible.

We thank the referee for finding our discovery that 7SK regulates epidermal differentiation reliable and convincing. As described below in detail, the main concern of this referee is using 4SU RNA sequencing data in splicing analyses. We agree with the referee's concern and omitted the splicing analyses using the 4SU datasets from the study. Instead, we have now using total RNA sequencing datasets to determine the splicing differences.

1) Figure 5 shows by isolating pulse labelled nascent RNA (4SU) that 7SK depletion does reduce nascent transcription levels. I think these data would benefit from more gene screen shots clearly showing this reduction in nascent transcription. The TP63 gene data is hard to appreciate (Fig 5B). Really it would be good to have back up data using GRO-seq or mNET-seq here.

As requested by the referee, we have now included more genome browser shots throughout our manuscript to better highlight the differences in nascent RNA production.

We also agree that mNET-seq would be the ideal method to further validate our data. We have spent a year in trying to establish the method according to the published protocols. Unfortunately, the antibodies used in the original protocol are no longer available, and we failed to get the method to work using alternative antibodies. We believe that GRO-seq would not give us more information as it follows the same principle than the 4SU-seq approach.

To accommodate the referee's comments, we have now re-analysed our current data sets in far greater detail and provide further data to substantiate our claims.

2) I have serious misgivings about some of transcriptomic analysis related to splicing aimed at understanding the mechanism behind the 7SK kd phenotypes. Principally I think 4SU labelling is problematic for looking at splicing as it is possible that this analogue incorporation into RNA may impair splicing which relies on splice site base pairing to snRNAs. Really these

data need repeating using other nascent transcription analysis methods that will give unbiased splicing efficiency measures. Possibly a comparison of chromatin vs nucleoplasm RNA looking for intron retention (+/-7SK depletion). Alternatively, mNET-seq using the splicing specific Pol II CTD S5P selection would be a good approach (see Nojima et al. Mol Cell 72,369 2018).

The referee raises a very good point. Although 4SU-labelled RNA has been used to study splicing effects before (Windhager et al., 2012), we agree that the 4SU-labelled RNA should probably not be used to identify splicing differences. Therefore, we omitted these analyses from the manuscript and now perform the splicing analyses only on total RNA-seq datasets.

We now also focus on the main function of 7SK on transcription initiation and provide more and novel mechanistic insights on how *Rn7sk* is required is to orchestrate bi-directional transcription. Moreover, we identify the cell cycle regulators as direct targets of *Rn7sk*-mediated transcriptional initiation.

3) I am unsure if the bioinformatic correlation of low GC content and weaker 3' splice sites for 7SK depletion repressed genes is a very meaningful correlation.

We agree with the referee and have now omitted the panels from the revised version of this manuscript.

4) I don't understand why PlaB treatment which will clearly block all splicing has an antagonistic effect on 7SK depletion induced splicing defects. Surely the combination of 7SK depletion followed by PlaB treatment should cause additive effects on splicing inhibition, not rescue effects? Is this again related to potential problems with using 4SU labelling to study splicing efficiency. Gene specific screen shots are needed here to present these data in a more convincing manner.

The referee is correct, PlaB treatment did indeed have an additive effect on splicing. It did however rescue *Rn7sk*-sensitive gene transcription because the targets were not degraded.

However, because the referee is correct in voicing concern over the use of 4SU data in splicing analyses (see also comment 2), we have re-analyzed our data using only the total RNA-seq data sets.

We do believe that splicing is affected in the *Rn7sk*-depleted cells, but deeper analyses into impaired transcription in the absence of *Rn7sk* now provided us with a novel mechanism explaining the cellular phenotype observed *in vitro* and *in vivo*. Therefore, we now focus our study on bi-directional transcription.

Overall, I feel that while the first parts of this paper look compelling and interesting I don't consider the splicing connection is convincingly made. More work is needed as suggested above to either confirm or rule out a splicing connection.

We agree with the referee and now provide extensive and new analyses explaining the direct role of the 7SK RNP complex in transcription initiation.

References

- Bazi, Z., Bertacchi, M., Abasi, M., Mohammadi-Yeganeh, S., Soleimani, M., Wagner, N., and Ghanbarian, H. (2018). Rn7SK small nuclear RNA is involved in neuronal differentiation. *J Cell Biochem* 119, 3174-3182.
- Castelo-Branco, G., Amaral, P.P., Engstrom, P.G., Robson, S.C., Marques, S.C., Bertone, P., and Kouzarides, T. (2013). The non-coding snRNA 7SK controls transcriptional termination, poising, and bidirectionality in embryonic stem cells. *Genome Biol* 14, R98.
- Core, L., and Adelman, K. (2019). Promoter-proximal pausing of RNA polymerase II: a nexus of gene regulation. *Genes & development* 33, 960-982.
- Creyghton, M.P., Cheng, A.W., Welstead, G.G., Kooistra, T., Carey, B.W., Steine, E.J., Hanna, J., Lodato, M.A., Frampton, G.M., Sharp, P.A., et al. (2010). Histone H3K27ac separates active from poised enhancers and predicts developmental state. *Proc Natl Acad Sci U S A* 107, 21931-21936.
- Flynn, R.A., Do, B.T., Rubin, A.J., Calo, E., Lee, B., Kuchelmeister, H., Rale, M., Chu, C., Kool, E.T., Wysocka, J., et al. (2016). 7SK-BAF axis controls pervasive transcription at enhancers. *Nat Struct Mol Biol* 23, 231-238.
- Gressel, S., Schwalb, B., and Cramer, P. (2019). The pause-initiation limit restricts transcription activation in human cells. *Nature communications* 10.
- Gressel, S., Schwalb, B., Decker, T.M., Qin, W., Leonhardt, H., Eick, D., and Cramer, P. (2017). CDK9-dependent RNA polymerase II pausing controls transcription initiation. *eLife* 6.
- Nguyen, V.T., Kiss, T., Michels, A.A., and Bensaude, O. (2001). 7SK small nuclear RNA binds to and inhibits the activity of CDK9/cyclin T complexes. *Nature* 414, 322-325.
- Shao, W.Q., and Zeitlinger, J. (2017). Paused RNA polymerase II inhibits new transcriptional initiation. *Nature genetics* 49, 1045-+.
- Studniarek, C., Tellier, M., Martin, P.G.P., Murphy, S., Kiss, T., and Egloff, S. (2021). The 7SK/P-TEFb snRNP controls ultraviolet radiation-induced transcriptional reprogramming. *Cell reports* 35, 108965.
- Windhager, L., Bonfert, T., Burger, K., Ruzsics, Z., Krebs, S., Kaufmann, S., Malterer, G., L'Hernault, A., Schilhabel, M., Schreiber, S., et al. (2012). Ultrashort and progressive 4sU-tagging reveals key characteristics of RNA processing at nucleotide resolution. *Genome Research* 22, 2031-2042.
- Yang, Z., Zhu, Q., Luo, K., and Zhou, Q. (2001). The 7SK small nuclear RNA inhibits the CDK9/cyclin T1 kinase to control transcription. *Nature* 414, 317-322.

REVIEWERS' COMMENTS

Reviewer #2 (Remarks to the Author):

The authors conducted new experiments and modified the text to address the critical points I raised in its first version.

However, I have one final comment about the manuscript: The authors generated two different transgenic mouse lines carrying floxed Rn7sk alleles (plus and minus PSE). It becomes not clear whether the in vivo studies were performed with both mouse lines or which line was involved performing a particular set of experiments. This should be stated in the figure legends and material/methods part of the manuscript.

Reviewer #3 (Remarks to the Author):

This extensively revised paper describes the effect of either a knock out or siRNA mediated depletion of 7sk snRNA, initially in skin epidermis differentiation. In the first part of the paper it is shown that skin epidermal division is perturbed by loss of 7sk resulting in a reduction in cellularity. The rest of the paper seeks to understand this epidermal phenotype by looking at the effect of 7sk depletion on gene expression (especially transcription and associated RNA processing).

7sk is well known to negatively regulate the key transcription elongation kinase Cdk9 as part of the cyclin T associated protein pTEFb. This regulation is achieved by 7sk sequestration into an inactive complex. The effect of 7sk loss might therefore have been expected to activate many genes. However, a class of genes was shown here to be inhibited by loss of 7sk. These inhibited genes functionally correlate with often divergently transcribed genes that are associated with cell cycle and chromosome organisation. This may account for the effect of 7sk depletion on epidermal cellularity.

Overall, I feel that this revised paper addresses my previous concerns (especially overclaiming effects on splicing). However, one is still left without a very clear mechanistic understanding of 7sk function in cellular differentiation. Rather the data presented raises interesting questions for further experimental work that is beyond the scope of this study.

NCOMMS-19-11141432A

In response to referees' comments, we have made the following changes to our manuscript.

REVIEWERS' COMMENTS

Reviewer #2:

The authors conducted new experiments and modified the text to address the critical points I raised in its first version.

However, I have one final comment about the manuscript: The authors generated two different transgenic mouse lines carrying floxed Rn7sk alleles (plus and minus PSE). It becomes not clear whether the in vivo studies were performed with both mouse lines or which line was involved performing a particular set of experiments. This should be stated in the figure legends and material/methods part of the manuscript.

Both transgenic lines were analysed and displayed the same phenotype. Most data in Figure 1 and Supplementary Figure 1 were obtained from line 1 (see Figure 1a). Exceptions were Figure 1e and f, which contain the pooled data from both lines. This information is now included in the Methods section and the respective Figure legends.

Reviewer #3:

This extensively revised paper describes the effect of either a knock out or siRNA mediated depletion of 7sk snRNA, initially in skin epidermis differentiation. In the first part of the paper it is shown that skin epidermal division is perturbed by loss of 7sk resulting in a reduction in cellularity. The rest of the paper seeks to understand this epidermal phenotype by looking at the effect of 7sk depletion on gene expression (especially transcription and associated RNA processing).

7sk is well known to negatively regulate the key transcription elongation kinase Cdk9 as part of the cyclin T associated protein pTEFb. This regulation is achieved by 7sk sequestration into an inactive complex. The effect of 7sk loss might therefore have been expected to activate many genes. However, a class of genes was shown here to be inhibited by loss of 7sk. These inhibited genes functionally correlate with often divergently transcribed genes that are associated with cell cycle and chromosome organisation. This may account for the effect of 7sk depletion on epidermal cellularity.

Overall, I feel that this revised paper addresses my previous concerns (especially overclaiming effects on splicing). However, one is still left without a very clear mechanistic understanding of 7sk function in cellular differentiation. Rather the data presented raises interesting questions for further experimental work that is beyond the scope of this study.

We thank the referee for finding our manuscript now suitable for publication.